# A²RBench: An Automatic Paradigm for Formally Verifiable Abstract Reasoning Benchmark Generation

Qingchuan Ma [1]   Yuexiao Ma [1]   Yongkang Xie [1]   Tianyu Xie [1]   Xiawu Zheng [1 2]   Rongrong Ji [1 2]

## Abstract

Abstract reasoning ability reflects the intelligence and generalization capacity of LLMs to extract and apply abstract rules. However, accurately measuring this ability remains challenging: existing benchmarks either rely on expensive manual annotation, limiting their scale, or risk measuring memorization rather than genuine reasoning. To address this, we introduce an automated pipeline named A²RBench, encompassing generation, expansion, evaluation, and analysis. Specifically, in the generation stage, LLMs create diverse tasks demanding genuine reasoning; in the expansion stage, LLMs reuse validated rules and expand new input spaces to generate task variations, achieving scaling. However, such a process may cause hallucinations. To eliminate it, we further establish a theoretical framework and prove that programmatic verification—testing whether the inverse operation perfectly reverses the forward operation (cycle consistency)—guarantees a unique solution. Through extensive evaluations on mainstream LLMs, we find: (1) Current LLMs exhibit fundamental deficiencies in abstract reasoning, with top models significantly underperforming humans on a representative subset (39.8% vs. 68.5%). (2) Current LLMs fall far short of 2D and 1D in the complexity of generated 3D tasks, revealing their lack of understanding of high-dimensional tasks. (3) Counterintuitively, inputs with higher information complexity can simplify the reasoning process. Code and data are available at: https://github.com/MAC-AutoML/A2Rbench.

[1]Key Laboratory of Multimedia Trusted Perception and Efficient Computing, Ministry of Education of China, Xiamen University, 361005, P.R. China. [2]Institute of Artificial Intelligence, Xiamen University. Correspondence to: Xiawu Zheng <zhengxiawu@xmu.edu.cn>.

*Proceedings of the 43rd International Conference on Machine Learning*, Seoul, South Korea. PMLR 306, 2026. Copyright 2026 by the author(s).

## 1. Introduction

Abstract reasoning, a cornerstone of intelligence (Hofstadter, 1995), reflects the deliberate "System 2" cognition (Kahneman, 2011) that separates genuine algorithmic inference from sophisticated pattern matching. For Large Language Models (LLMs), which have been trained to follow human instructions through techniques like reinforcement learning from human feedback (Ouyang et al., 2022) and are primarily built upon the Transformer architecture (Vaswani et al., 2017), the ability to perform such reasoning is a critical frontier. While techniques like Chain-of-Thought prompting (Wei et al., 2022)—even those bolstered by high-quality, step-by-step supervision (Yeo et al., 2025)—have elicited impressive multi-step reasoning capabilities, a fundamental question remains: are these models truly reasoning, or simply performing sophisticated pattern matching? This concern is captured by the "stochastic parrots" critique (Bender et al., 2021), which warns that models may blur the line between genuine understanding and high-fidelity mimicry. Thus, answering this challenge requires evaluation tools that demand genuine reasoning, offer diversity, and enable scalability.

However, existing benchmarks face a fundamental trade-off between reasoning demands and scalability. Manually designed benchmarks like the Abstraction and Reasoning Corpus (ARC) require genuine reasoning but remain limited in scale (Chollet, 2019). Conversely, large-scale datasets such as GSM8K, MATH, and BIG-bench, while scalable, risk measuring rote memorization rather than genuine reasoning (Cobbe et al., 2021; Hendrycks et al., 2021; Srivastava et al., 2023).

To transcend this impasse, we introduce an automated pipeline named A²RBench, encompassing generation, expansion, evaluation, and analysis. Specifically, in the generation stage, LLMs create diverse tasks demanding genuine reasoning; in the expansion stage, LLMs reuse validated rules and expand new input spaces to generate task variations, achieving scaling. However, such a process may cause hallucinations. To eliminate it, we establish a theoretical framework where each task is implemented as a pair of executable functions: a forward function $f$ and its inverse $g$. We prove that programmatic verifica-

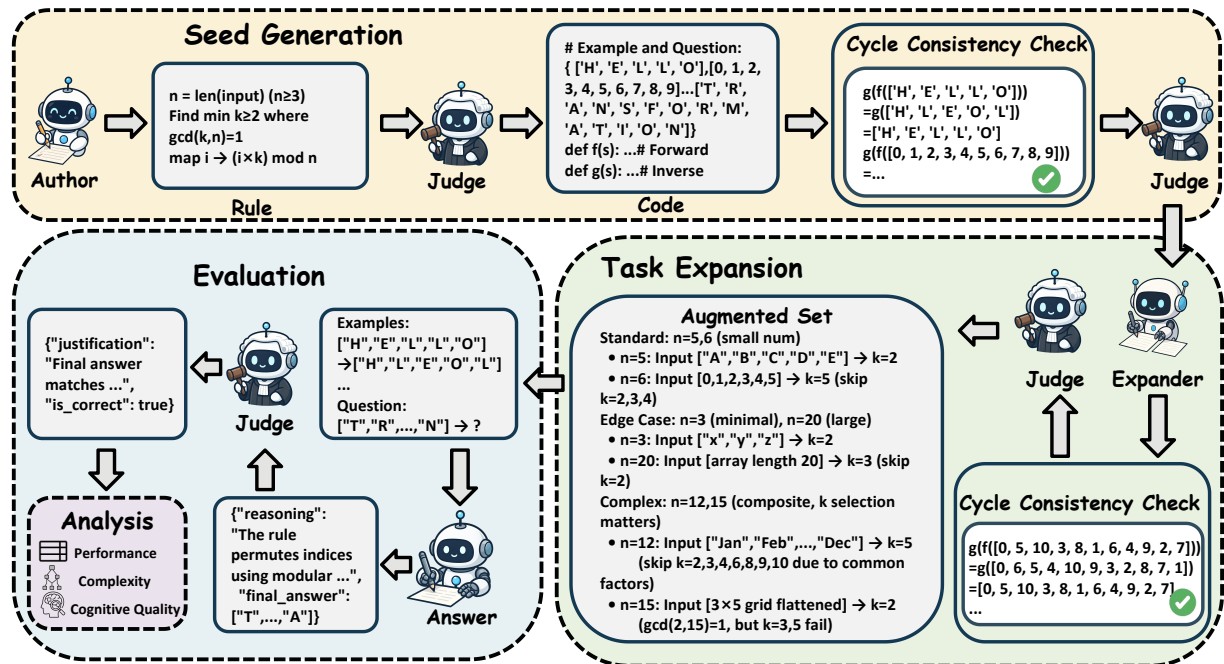

*Figure 1.* The A$^2$RBench automated pipeline illustrated with an example (rule: "index permutation via modular arithmetic"). The rule permutes positions: given input length $n$, find smallest integer $k$ where $2 \leq k \leq n-1$ and $\gcd(k, n) = 1$, then map position $i$ to $(i \times k) \mod n$. With $n = 14$, smallest coprime is $k = 3$, mapping $0 \rightarrow 0$, $1 \rightarrow 3$, $2 \rightarrow 6$, etc. **Stage 1 (Seed Generation):** Author model generates rule description validated by Judge model. Author implements forward function $f$ (encoder), inverse function $g$ (decoder), example inputs and query input in Python. Cycle Consistency Check verifies $g(f(x)) = x$ for all inputs, followed by Judge filtering of trivial cases. **Stage 2 (Task Expansion):** Expander model generates input variations across three difficulty levels (Standard, Edge Case, Complex), reusing validated rule code. Each variation undergoes cycle consistency and judge validation. **Stage 3 (Evaluation):** Solver models receive examples, infer rule, and answer queries. Judge model verifies correctness against ground truth; for symbolic tasks, symbol remapping ($\phi$) measures reliance on familiar tokens. **Stage 4 (Analysis):** We analyze across three dimensions: performance statistics (accuracy metrics), code complexity (AST-based difficulty), and cognitive quality (reasoning classification). This pipeline generates formally verified tasks at scale while enabling diagnostic evaluation beyond binary correctness.

tion—testing whether the inverse perfectly reverses the forward operation through cycle consistency check $g(f(x)) = x$—guarantees a unique solution (Theorem 3.1). This automated pipeline[1] produces formally verified problems that demand genuine reasoning. Moreover, the known ground-truth rules enable interpretability for analyzing model reasoning behavior.

Through extensive evaluations on mainstream LLMs, we uncover three key findings. First, current LLMs exhibit fundamental deficiencies in abstract reasoning, with the top-performing model achieving only 39.8% accuracy on a representative subset, significantly underperforming humans at 68.5%. Second, current LLMs fall far short of 2D and 1D in the complexity of generated 3D tasks, revealing their lack of understanding of high-dimensional tasks. Third, counterintuitively, inputs with higher information

complexity can simplify the reasoning process.

Specifically, our contributions are as follows:

- We establish a mathematical framework proving that cycle consistency ($g(f(x)) = x$) is sufficient to guarantee solution uniqueness, providing a formal foundation for automated verification without human judgment.

- We construct an automated pipeline that generates, expands, evaluates, and analyzes abstract reasoning tasks at scale.

- We conduct extensive evaluations on 14 mainstream models with fine-grained cognitive analysis, revealing fundamental limitations in abstraction capabilities and providing diagnostic insights beyond surface-level accuracy.

---

[1]The pipeline is automated after initialization with 20 manually curated seed rules from ARC (Chollet, 2019). All subsequent generation, expansion, and analysis stages require no human intervention.

## 2. Related Work

Abstract reasoning evaluation has evolved from psychometric instruments like Raven's Progressive Matrices (Raven, 1983) to computational benchmarks (Barrett et al., 2018), culminating in the Abstraction and Reasoning Corpus (ARC) (Chollet, 2019), though models struggle with concept generalization (Moskvichev et al., 2023). While large-scale datasets like GSM8K and BIG-bench (Cobbe et al., 2021; Srivastava et al., 2023) offer scalability, they risk measuring memorization over genuine "System 2" reasoning (Kahneman, 2011; Bonnefon & Rahwan, 2020). Recent work has begun disentangling these effects through symbolic remapping (Ma et al., 2025; Mirzadeh et al., 2024), yet the tension between cognitive depth and scalable validation persists.

This has catalyzed LLM-driven paradigms for data generation and evaluation. Self-improvement approaches—from self-instruction (Wang et al., 2023) to self-correction via reinforcement learning (Huang et al., 2023; Kumar et al., 2024)—have matured into self-play frameworks where models enhance reasoning through adversarial games (Cheng et al., 2024; Zhang et al.), self-generated rewards (Yuan et al., 2024), or generator-solver roles (Liu et al., 2025b), with applications spanning code generation (Lin et al., 2025) to theorem proving (Dong & Ma, 2025). Concurrently, program-aided prompting (Chen et al., 2022; Gao et al., 2023) leverages code for reasoning tasks. The LLM-as-a-judge paradigm enables scalable evaluation as automated graders (Zheng et al., 2023), red-team testers (Perez et al., 2022), bug detectors (McAleese et al., 2024), and benchmark generators (Bai et al., 2023; Muhlgay et al., 2024; Li et al., 2023; Desmond et al., 2024).

However, LLM-driven approaches inherit model unreliability. LLM judges suffer from subjectivity and bias (Shankar et al., 2024; Zhang et al., 2023), fundamentally lacking ground truth guarantees. Unlike step-by-step process supervision (Lightman et al., 2023) or execution feedback in self-play (Dong et al., 2024), we synthesize LLMs' generative power with deterministic code execution as a formal, one-time guarantee of logical integrity—ensuring both scalability and rigorous verification absent in prior automated approaches.

## 3. Theoretical Framework

Constructing an LLM-based benchmark for abstract reasoning faces two core challenges. First, generative hallucination may produce logically flawed tasks, requiring robust automated filtering. Second, models may reach correct answers via superficial pattern matching rather than genuine rule induction, necessitating a framework to assess reasoning quality. This section establishes the mathematical foundation addressing both challenges.

### 3.1. Formalizing Abstract Reasoning

Abstract reasoning—the ability to extract generalizable patterns from concrete instances and apply them to novel situations—is a cornerstone of human cognition (Penn et al., 2008; Holyoak & Morrison, 2012). Building upon this cognitive foundation, recent work has formalized this process as a two-stage computational procedure (Ma et al., 2025). In the first stage, Abstraction extracts a general rule $r$ from a set of provided examples $\mathcal{E}$. In the second stage, Reasoning applies this rule to a query input $x_q$ to produce the output $y_q$.

To illustrate this process concretely, consider observing that "a poodle barks," and "a bulldog barks." The abstraction stage requires extracting the general rule—"dogs bark"—from these specific instances. The reasoning stage then applies this rule to infer that a newly encountered beagle will also bark. This two-stage decomposition captures the essence of moving from concrete observations to abstract principles and back to concrete predictions.

But how do we measure this capability computationally in a benchmark setting? A typical abstract reasoning task presents a model with a set of example transformations, challenges it to infer the underlying rule, and then apply this rule to a new input. Formally, such a task consists of:

- A set of example pairs $\mathcal{E} = \{(x_i, y_i)\}_{i=1}^k$, where $x_i \in \mathcal{X}$ and $y_i \in \mathcal{Y}$

- A query input $x_q \in \mathcal{X}$

- An underlying rule $r$ such that $y_i = r(x_i)$ for all examples

The spaces $\mathcal{X}$ and $\mathcal{Y}$ represent discrete data structures, including 1D sequences, 2D grids, and 3D voxel arrays in our benchmark.

To rigorously evaluate whether an LLM possesses this capability, we must formalize what it means to "solve" such a task. Given the example set $\mathcal{E}$ and the query $x_q$, a solver model must explicitly perform both stages:

$$\hat{r} \leftarrow \mathcal{M}_{\text{solver}}(\mathcal{E}) \quad \text{(Stage 1: Infer the rule)}, \quad (1)$$

$$\hat{y}_q = \hat{r}(x_q) \quad \text{(Stage 2: Apply the rule)}. \quad (2)$$

Critically, our evaluation protocol requires solver models to explicitly articulate their inferred rule $\hat{r}$ and show their reasoning steps. This verbalization enables diagnostic analysis of reasoning quality (Section 3.3), allowing us to distinguish genuine rule induction from superficial pattern matching.

As formalized above, a fundamental requirement for building such a benchmark emerges: all examples in $\mathcal{E}$ and the final question-answer pair $(x_q, y_q)$ must follow the same underlying rule $r$. However, checking semantic equivalence between a natural language description and a set of input-output pairs requires either expensive human judgment or another LLM evaluator, reintroducing the very unreliability (hallucination, subjectivity) we seek to avoid. To address this challenge, we make a critical design choice: we mandate that the LLM generates the rule $r$ as executable Python code, along with the input variables (example inputs $\{x_i\}$ and query input $x_q$). The corresponding outputs $\{y_i\}$ and the ground-truth answer $y_q$ are then derived not from the LLM's generation, but by programmatically executing the rule function $r$ on these inputs:

$$y_i := r(x_i), \ \forall i \in \{1, \ldots, k\}, \quad y_q := r(x_q). \quad (3)$$

This transformation into deterministic programs ensures perfect consistency between examples and the underlying rule by construction, enabling automated verification and scalable development of abstract reasoning benchmarks. Nevertheless, programmatic execution alone cannot detect logical errors in the LLM-generated rule code. This remaining challenge necessitates a rigorous verification mechanism, which we establish in the following section.

### 3.2. Ensuring Task Validity via Cycle Consistency

The code-based design in Section 3.1 ensures consistency between examples and the rule by construction. However, a critical challenge remains: the LLM-generated code itself may contain logical errors due to hallucination. A function may execute without runtime errors yet embody fundamentally flawed logic. This necessitates a robust, programmatic filtering mechanism to detect and eliminate such invalid tasks.

Our approach is to leverage mathematical structure. Rather than relying on another LLM to subjectively judge code quality—which would reintroduce the very unreliability we seek to avoid—we ask: what mathematical properties must a valid reasoning task possess? If we can formalize these properties, we can design an automated check that either proves validity or detects flaws with certainty. This motivates a principled definition of task validity.

To establish such a mechanism, we first require a formal definition of what constitutes a "valid task." Drawing inspiration from the mathematical concept of a Well-Posed Problem (Hadamard, 1902), we define a well-posed abstract reasoning task as one that satisfies three criteria:

1. **Uniqueness**: Every input has exactly one correct output, ensuring an unambiguous answer.

2. **Consistency**: The same rule applies uniformly to all examples and the query.

3. **Verifiability**: Logical soundness can be confirmed via a deterministic, automated procedure.

With this definition established, we now turn to a key question: what mathematical properties must a rule function $r$ possess to satisfy these three criteria? The answer lies in analyzing the type of mapping from inputs $\mathcal{X}$ to outputs $\mathcal{Y}$. Each criterion imposes strict constraints that progressively narrow the space of permissible functions:

**Uniqueness eliminates One-to-Many mappings.** If a single input can map to multiple outputs, the correct answer becomes inherently ambiguous, violating the first criterion (see Appendix A.2.1 for concrete examples).

**Verifiability eliminates Many-to-One mappings.** While deterministic, such mappings lose information—multiple inputs collapse to the same output. This irreversibility makes it impossible to construct an inverse operation for automated verification (Appendix A.2.2).

**This leaves One-to-One (Bijective) mappings as a viable design choice for automatic verification.** A bijective function guarantees that every input maps to a unique output (satisfying Uniqueness), and crucially, it is invertible—there exists an inverse function $g$ such that $g(f(x)) = x$. This two-way reversibility provides the mathematical foundation for our automated validation framework.

We therefore enforce that all generated rules must be bijections. This restriction allows us to design the Cycle Consistency Check: we require the Author LLM to generate both a forward function $f$ (encoder) and its inverse $g$ (decoder), then programmatically verify that the round-trip transformation returns to the original input. Formally, for all $x$ in the input space $\mathcal{X}$:

$$g(f(x)) = x, \quad \forall x \in \mathcal{X} \quad (4)$$

This automated test serves as a powerful filter in practice, successfully detecting flawed logic, implementation bugs, and inconsistencies between forward and inverse functions (concrete examples in Appendix A.3). However, a fundamental question remains: does passing this check guarantee that a task is well-posed, or is it merely a useful heuristic? To provide a rigorous foundation for our automated pipeline, we prove the former:

**Theorem 3.1** (Cycle Consistency Guarantees Validity). *If a generated task, defined by functions $(f, g)$, satisfies the cycle consistency check (Eq. 4), it is guaranteed to have a unique solution and be logically verifiable, thereby qualifying as a Well-Posed Problem.*

The proof is provided in Appendix A.1. This theorem forms the theoretical foundation of our automated filtering pipeline, ensuring that our generated tasks are guaranteed by both theoretical well-posedness and programmatic logical correctness.

### 3.3. Distinguishing Reasoning from Surface Fitting

The framework in Section 3.2 guarantees that each task has a unique, valid solution—ensuring the structural integrity of our benchmark. However, structural validity alone is insufficient for evaluating abstract reasoning. Recall that our core objective, as motivated in the introduction, is to distinguish genuine rule induction from sophisticated pattern matching—the difference between true "System 2" reasoning and the "stochastic parrot" behavior (Bender et al., 2021). A model might produce the correct answer through multiple pathways: by inferring the intended general rule, by overfitting to examples, or even by lucky guessing. To achieve our diagnostic goal, we must look beyond binary correctness and analyze the quality of the reasoning process itself.

Consider the sequence transformations $[1, 2] \rightarrow [2, 4]$ and $[3, 4] \rightarrow [6, 8]$. For the input $[5, 6]$, a model might infer the intended rule ("double each element"), an overly specific variant ("multiply the first element by 2, then the second by 2"), or a complex polynomial overfitted to the examples—all yielding the correct answer $[10, 12]$, yet only the first exhibits true generalization. This illustrates a fundamental ambiguity: for any example set, multiple rules can perfectly explain the observations, and the inferred rule $\hat{r}$ may fit all examples flawlessly yet differ fundamentally from the ground-truth rule $r$.

Which rule should be considered "correct"? We appeal to Occam's Razor: when multiple explanations fit the data equally well, the simplest one is most likely correct. This principle can be connected to Solomonoff's theory of inductive inference (Solomonoff, 1964): given the set of all rule hypotheses $\mathcal{C}_R(\mathcal{E})$ that are consistent with a set of examples $\mathcal{E}$, the simplest consistent rule can be written as

$$r^* = \arg \min_{r \in \mathcal{C}_R(\mathcal{E})} K(r) \tag{5}$$

where $K(r)$ denotes Kolmogorov complexity. While Kolmogorov complexity is uncomputable, Eq. 5 serves as an explanatory criterion for evaluating reasoning quality. To operationalize this, we leverage that our evaluation protocol (Section 3.1) requires models to explicitly articulate their inferred rule $\hat{r}$ and reasoning steps. We employ an Analyst LLM as a heuristic proxy to classify the quality of correct answers:

- **Surface Fitting**: The model produces the correct answer through flawed logic or pattern matching, without articulating a valid general rule.

- **Inferior Rule**: The model identifies a rule that works but is unnecessarily complicated or overly specific.

- **True Generalization**: The model identifies the simplest, most general rule.

This cognitive analysis allows our benchmark to measure reasoning quality, not merely final answer accuracy, providing deeper diagnostic insights into abstract reasoning in LLMs. We validate this Analyst LLM classification on stratified human-annotated subsets, with details reported in Appendix D.2.

## 4. Methodology

To operationalize the framework from Section 3, we design a four-stage automated pipeline (Figure 1): seed generation, task expansion, evaluation, and analysis.

### 4.1. Seed Generation

To construct our benchmark, we first generate a small set of high-quality seed problems, each representing a distinct abstract rule. These seeds will later be expanded into many variations at low cost.

Our seeds span three logical dimensions and two reasoning domains. The three dimensions are: 1D (sequences like $[1, 2, 3]$), 2D (grids like $[[a, b], [c, d]]$), and 3D (voxel cubes). The two domains are: symbolic rules (structural transformations independent of symbol meaning) and semantic rules (transformations requiring external knowledge).

We generate each seed through a two-stage process. First, an author model $\mathcal{M}_{\text{author}}$ generates a natural language description of a bijective rule and its inverse, $(D_f, D_g)$, guided by randomly sampled inspiration rules from ARC (Chollet, 2019):

$$(D_f, D_g) = \mathcal{M}_{\text{author}}(\mathcal{P}_{\text{rule}}(R_{\text{inspire}})) \tag{6}$$

A judge model $\mathcal{M}_{\text{judge}}$ performs a preliminary review to filter out descriptions with obvious logical flaws. If approved, the author model then implements the rule as executable Python code:

$$S_{\text{code}} = (f, g, \mathcal{X}) = \mathcal{M}_{\text{author}}(\mathcal{P}_{\text{code}}(D_f, D_g)) \tag{7}$$

where $S_{\text{code}}$ contains the forward function $f$, inverse function $g$, and input set $\mathcal{X}$ (examples and query). We adopt this two-stage design because natural language descriptions are easier for the judge to evaluate than code, allowing us to discard many invalid concepts before expensive code generation.

To ensure each seed is well-posed per Theorem 3.1, we programmatically execute the Cycle Consistency Check, verifying that $g(f(x)) = x$ for all inputs. The judge model then performs a final review to discard trivial cases (e.g., identity functions or empty operations). This multi-step validation ensures logical soundness but makes seed generation costly, motivating the efficient expansion strategy in the next section.

### 4.2. Task Expansion

To scale the benchmark efficiently, we expand each validated seed into multiple variations. The key insight is to reuse the seed's verified rule and code, generating only new input spaces. This allows us to create diverse test cases while maintaining the formal guarantees established during seed generation.

An expander model $\mathcal{M}_{\text{expander}}$ generates new input sets $\mathcal{X}_i$ for each seed, where each input set contains new example inputs and a new query input:

$$\mathcal{X}_i = \mathcal{M}_{\text{expander}}(\mathcal{P}_{\text{expand}}(D_f, S_{\text{code}}, H_x)) \qquad (8)$$

Here, $\mathcal{P}_{\text{expand}}$ is the prompt template providing the rule description $D_f$, the reference code $S_{\text{code}}$, and the history of previously generated inputs $H_x$ to ensure diversity.

We generate up to 9 variations for each seed, following a three-phase strategy designed to systematically probe different aspects of solver capabilities. Variations V1-V3 focus on standard cases to verify baseline understanding of the rule. Variations V4-V6 target edge cases (e.g., empty structures, boundary values) to test robustness. Variations V7-V9 introduce complex or adversarial patterns to stress-test reasoning under challenging conditions. This phased approach ensures comprehensive coverage of the problem space.

Each generated variation undergoes the same validation as the seeds: we execute the code to obtain ground-truth outputs, verify cycle consistency, and use the judge model to filter out trivial or flawed instances. This reuse-and-expand strategy achieves remarkable economic efficiency: while seed generation costs $0.19 per task, expansion costs merely $0.005 per task—a $38\times$ reduction. As detailed in Appendix C.4 and Table 3, this automated paradigm reduces per-task cost to $0.016 on average, orders of magnitude lower than manually curated benchmarks.

### 4.3. Evaluation Protocol

We evaluate solver models by comparing their outputs against ground-truth answers using a judge model $\mathcal{M}_{\text{judge}}$. Beyond measuring accuracy, we also assess whether models rely on familiar symbols rather than abstract structure.

Since our benchmark includes symbolic reasoning tasks,

we adopt the symbolic dependency metric (Ma et al., 2025). We first measure a model's accuracy on the original task set, denoted as $\text{Acc}_{\text{P0}}$. Then, for symbolic tasks only (as semantic tasks inherently depend on symbol meaning, e.g., chemical elements, alphabetical order), we apply a symbol mapping function $\phi$ that remaps every token (e.g., '0' $\rightarrow$ 'a', '1' $\rightarrow$ 'b') while preserving the underlying logical structure. A model that depends on familiar symbols will fail when symbols change, while a model that infers the true rule will maintain performance. We measure accuracy on these remapped tasks as $\text{Acc}_{\text{P1}}$. The symbolic dependency score is simply the performance drop:

$$\Delta_S = \text{Acc}_{\text{P0}} - \text{Acc}_{\text{P1}} \qquad (9)$$

A larger $\Delta_S$ indicates greater reliance on familiar symbols.

### 4.4. Analysis

Traditional benchmarks measure only final answer correctness. We aim to understand two deeper questions: how difficult is the task, and how does the model reason? Our framework enables this because every task is code (allowing objective difficulty measurement) and the ground-truth rule is known (allowing reasoning quality assessment).

We perform three analyses. First, we compute performance statistics—accuracy across rule types and dimensions—to establish what happened. Second, we analyze the Abstract Syntax Tree (AST) of $S_{\text{code}}$ to measure task complexity via metrics like loop depth and conditional branches, revealing how hard the task is. Third, we use analyst model $\mathcal{M}_{\text{analyst}}$ to classify the solver's chain-of-thought as surface fitting, inferior rule, or true generalization according to the Occam-style criterion in Eq. 5, diagnosing why the model succeeded or failed. These three perspectives jointly answer: what is the result, how difficult is the task, and why.

## 5. Results and Analysis

Our evaluations expose three critical boundaries of current LLMs' reasoning abilities. First, current LLMs demonstrate fundamental deficiencies in abstract reasoning, significantly underperforming humans. Second, current LLMs fall far short of 2D and 1D in the complexity of generated 3D tasks, revealing their lack of understanding of high-dimensional tasks. Third, counterintuitively, inputs with higher information complexity can simplify the reasoning process.

### 5.1. Reasoning Limitations

Current LLMs show significant deficiencies in abstract reasoning. As shown in Table 1, the top-performing model, Gemini3-Pro, achieves only 40.9% overall accuracy. On a representative subset, models perform much worse than

*Table 1.* **Main Results on A²RBench.** Model performance across three dimensions. **Overall Performance:** *Total/Sym/Sem Acc* denote accuracy on all 1,054 tasks, symbolic tasks (structure-based), and semantic tasks (knowledge-dependent), respectively. **Symbolic Dependency:** *Original (P0)* and *Mapped (P1)* measure accuracy on symbolic tasks before and after symbol remapping (e.g., '0' → 'a'); *Gap ($\Delta_S$)* represents the performance difference before and after remapping, quantifying reliance on familiar symbols—larger gap indicates stronger symbolic dependency. **Generalization Gap:** *Seed (V0)* and *Aug. (V1-9)* show accuracy on original seed tasks versus augmented variations (standard/edge/adversarial cases).

| MODEL | OVERALL PERFORMANCE | | | SYMBOLIC DEPENDENCY ANALYSIS | | | GENERALIZATION GAP | |
|---|---|---|---|---|---|---|---|---|
| | TOTAL ACC | SYM ACC | SEM ACC | ORIGINAL (P0) | MAPPED (P1) | GAP ($\Delta_S$) | SEED (V0) | AUG. (V1-9) |
| **GEMINI3-PRO** (GOOGLE, 2025) | **40.9%** | **37.0%** | 48.6% | 39.3% | **34.8%** | 4.6% | **39.8%** | **41.0%** |
| GPT-5 (SINGH ET AL., 2025) | 39.0% | 32.5% | **52.0%** | **41.3%** | 23.6% | **17.7%** | 38.9% | 39.0% |
| GPT-5-MINI (SINGH ET AL., 2025) | 36.9% | 31.5% | 47.7% | 40.2% | 22.8% | 17.4% | 34.3% | 37.2% |
| O4-MINI (OPENAI, 2025) | 33.7% | 28.3% | 44.3% | 36.5% | 20.2% | 16.2% | 34.3% | 33.6% |
| CLAUDE-SONNET-4.5 (ANTHROPIC, 2025) | 30.6% | 33.3% | 25.3% | 34.2% | 32.5% | 1.7% | 23.1% | 31.5% |
| GPT-5.2 (SINGH ET AL., 2025) | 28.3% | 30.5% | 23.9% | 32.5% | 28.5% | 4.0% | 21.3% | 29.1% |
| GEMINI-2.5-FLASH (COMANICI ET AL., 2025) | 27.7% | 26.4% | 30.4% | 29.3% | 23.4% | 6.0% | 25.0% | 28.0% |
| GLM-4.6 (AI, 2025) | 21.0% | 23.1% | 16.8% | 24.8% | 21.4% | 3.4% | 13.9% | 21.8% |
| DEEPSEEK-V3.2 (LIU ET AL., 2025A) | 20.8% | 20.8% | 20.7% | 23.4% | 18.2% | 5.1% | 17.6% | 21.1% |
| QWEN3-32B (YANG ET AL., 2025) | 19.3% | 22.5% | 12.8% | 23.1% | 21.9% | 1.1% | 6.5% | 20.7% |
| GPT4O-MINI (OPENAI, 2024) | 16.1% | 19.5% | 9.4% | 23.9% | 15.1% | 8.8% | 8.3% | 17.0% |
| QWEN3-14B (YANG ET AL., 2025) | 14.2% | 17.0% | 8.8% | 16.8% | 17.1% | -0.3% | 5.6% | 15.2% |
| GEMINI3-FLASH (GOOGLE, 2025) | 13.1% | 13.0% | 13.4% | 15.7% | 10.3% | 5.4% | 11.1% | 13.3% |
| QWEN3-8B (YANG ET AL., 2025) | 0.0% | 0.0% | 0.0% | 0.0% | 0.0% | 0.0% | 0.0% | 0.0% |

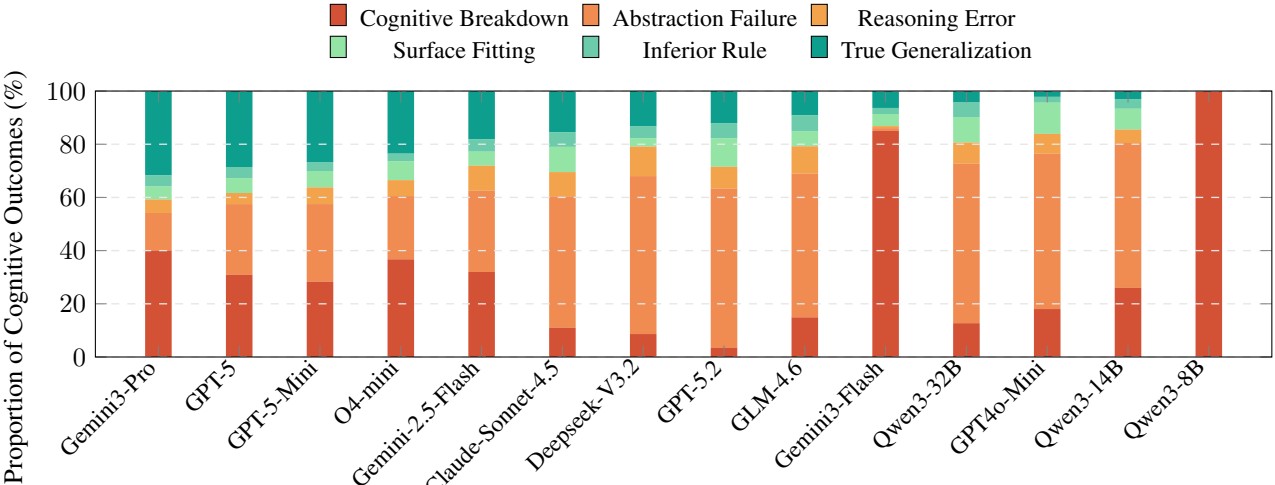

*Figure 2.* **A Spectrum of Cognitive Outcomes across LLMs.** This stacked bar chart categorizes each model's performance into a six-part spectrum. A warm color gradient (red to amber) denotes different types of failures, from fundamental breakdowns to execution errors. A cool color gradient (light green to dark green) distinguishes between three qualities of success: Surface Fitting, Inferior Rule, and True Generalization.

humans: 39.8% versus 68.5% (Appendix Table 11).

The poor performance stems from abstraction failure, not execution errors. Figure 2 shows that Abstraction Failure is the dominant error type across all models. Moreover, even when models produce correct answers, many rely on Surface Fitting or use an Inferior Rule rather than true generalization. This means abstraction failures cause both wrong answers and flawed reasoning behind correct ones.

Symbol remapping exposes this weakness further. The Symbolic Dependency gap ($\Delta_S$) in Table 1 quantifies the effect: models like GPT-5 drop 17.7% on remapped tasks, confirming over-reliance on familiar symbols rather than abstract structure. This weakness appears in both solving

and generating tasks.

### 5.2. Dimensionality Bottleneck

Author models struggle with higher-dimensional tasks, which affects the difficulty of generated problems. This leads to an unexpected pattern: all solver models consistently perform worse on 2D tasks than on 3D tasks, creating a 1D > 3D > 2D hierarchy (Figure 3). This V-shaped curve reveals a capability ceiling in the author models, not the intrinsic difficulty of 3D reasoning.

To understand this, we analyzed the generated code complexity. Appendix Table 9 shows that 2D tasks contain deeper conditional logic than 3D tasks—for example, O4-

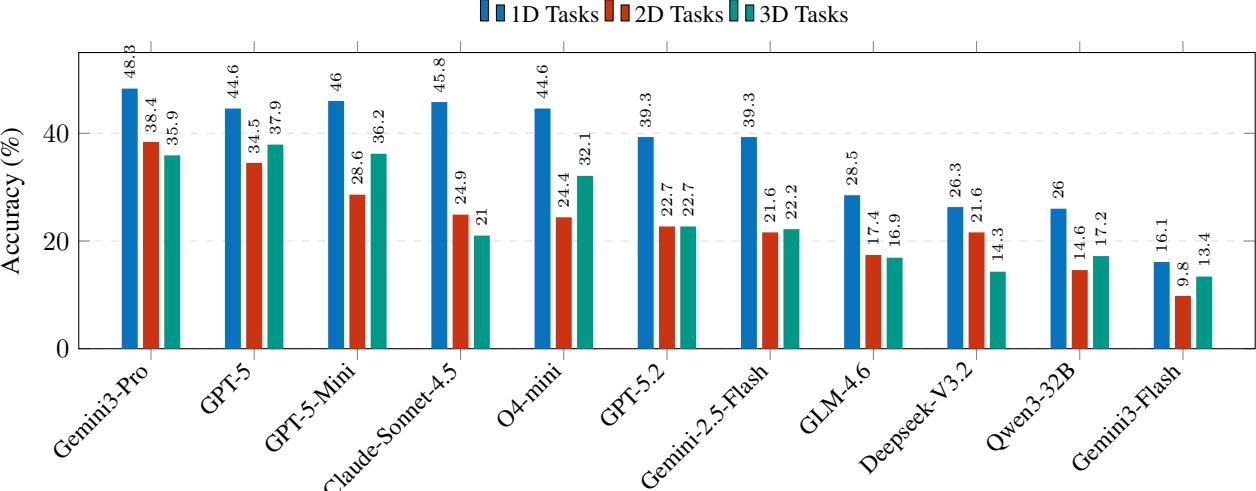

*Figure 3.* **Model Performance Across Logical Dimensions.** Accuracy of LLMs on 1D, 2D, and 3D abstract reasoning tasks. A consistent performance dip on 2D tasks (red bars) is observable across most models.

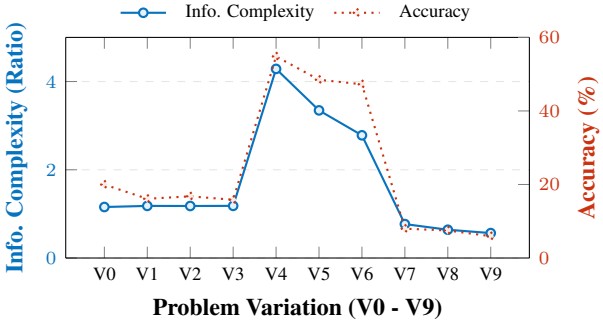

*Figure 4.* **The Augmentation Paradox.** A multi-axis plot correlating Information Complexity (blue) and Model Accuracy (red) across problem variations.

mini's 2D tasks have a Nested If Depth of 2.33 versus 1.40 for 3D. This suggests author models hit a cognitive trade-off: when handling 3D spatial complexity, they must simplify the underlying logic to maintain validity. This forced simplification makes 3D tasks accidentally easier for solvers.

Beyond dimensionality, task expansion revealed another unexpected pattern in how input structure affects reasoning difficulty.

### 5.3. Augmentation Paradox

Our findings reveal an unexpected pattern: certain generated variations proved paradoxically easier to solve despite higher input complexity. Figure 4 shows this relationship clearly. We measured input complexity using compression ratio—higher ratios indicate more structured, less compressible inputs. The figure reveals a striking correlation:

as input complexity peaks, model accuracy rises sharply.

The pattern is clearest at variation V4, where input complexity peaks at 4.29 yet accuracy jumps to 54.8%—nearly triple the baseline. Appendix Table 10 corroborates this via failure entropy (Shannon entropy over incorrect output distributions): V4 exhibits the lowest entropy (1.53 bits), indicating consistent rather than random errors. This suggests that highly structured inputs constrain the space of plausible rules, facilitating correct rule identification.

The mechanism is straightforward: highly structured inputs reduce ambiguity. They provide strong cues about the underlying transformation, limiting the number of rules that fit the examples. This finding confirms a key principle for benchmark design—true reasoning difficulty comes from rule ambiguity, not surface complexity.

### 5.4. Limitations and Future Work

While our framework ensures verifiability, its limitations suggest clear future directions. Generated task complexity is bounded by the author LLM's generative ceiling, addressable by employing stronger models for more diverse, sophisticated rules. Our analysis relies on an analyst LLM to approximate Occam's Razor, which could be replaced by more direct metrics for reasoning quality—potentially via mechanistic interpretability of internal model states (Venhoff et al., 2025). Beyond these refinements, the code-verification paradigm extends to other domains and provides a controlled testbed for investigating foundational phenomena. Extending the framework beyond bijective rules to many-to-one reasoning tasks is an important future direction. Preliminary fine-tuning results in Appendix D.3 further suggest that A²RBench may also serve as a train-

ing signal for answer convergence, output alignment, and discrete rule discrimination.

# 6. Conclusion

We tackled the challenge of scalable abstract reasoning evaluation through an automated paradigm combining LLM generation with code verification. To address generative hallucination, we established a theoretical framework proving that cycle consistency ($g(f(x)) = x$) guarantees problem well-posedness, enabling deterministic validation of logical soundness. Building on this foundation, we implemented a automated pipeline that generates, verifies, expands, evaluates, and analyzes tasks, yielding A$^2$RBench. Our work establishes a rigorous paradigm for scalable, formally verified cognitive evaluation of LLMs.

# Impact Statement

This paper presents work whose goal is to advance the field of Machine Learning. There are many potential societal consequences of our work, none which we feel must be specifically highlighted here.

# Acknowledgements

This work is supported by the National Key Research and Development Program of China (No. 2025YFE0113500), the National Natural Science Foundation of China (No. 62576299), and the Fundamental Research Funds for the Central Universities.

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

# A. Proofs and Supplementary Theory

## A.1. Proof of Theorem 3.1

The theorem asserts that if a task defined by a pair of generated functions, $(f, g)$, satisfies the cycle consistency property, then it is a well-posed problem. We must demonstrate that this satisfaction implies the three criteria for well-posedness: uniqueness, consistency, and verifiability.

Let the forward function be $f : \mathcal{X} \to \mathcal{Y}$ and the inverse function be $g : \mathcal{Y} \to \mathcal{X}$, where $\mathcal{X}$ and $\mathcal{Y}$ are the spaces of task inputs and outputs, respectively. The cycle consistency check programmatically verifies that for a set of test inputs $x \in \mathcal{X}_{\text{test}} \subset \mathcal{X}$, the following holds:

$$g(f(x)) = x \tag{10}$$

This establishes that the composition $g \circ f$ is the identity map on the tested subset of $\mathcal{X}$. Our generation framework assumes this property holds for the entire domain $\mathcal{X}$. We now prove each condition for well-posedness. Strictly speaking, cycle consistency establishes injectivity of $f$ on the task-defined input space; together with the generation process, which evaluates outputs in the image of $f$, the resulting invertibility claim is over this image set rather than over an arbitrary unrestricted codomain.

1. *Proof of Uniqueness:* A task has a unique solution if the function $f$ maps every input $x \in \mathcal{X}$ to exactly one output $y \in \mathcal{Y}$. This is guaranteed if $f$ is a well-defined function. The more rigorous aspect of uniqueness in our context is ensuring that no two distinct inputs map to the same output, which would make the rule ambiguous from an inverse perspective. The cycle consistency condition $g(f(x)) = x$ is sufficient to prove that $f$ is injective (one-to-one). To show this, assume for any two inputs $x_1, x_2 \in \mathcal{X}$ that $f(x_1) = f(x_2)$. Applying the function $g$ to both sides gives $g(f(x_1)) = g(f(x_2))$. By the cycle consistency property, this simplifies to $x_1 = x_2$. Thus, $f(x_1) = f(x_2)$ implies $x_1 = x_2$, which is the definition of injectivity. An injective function ensures that for any given input $x_q$, its output $y_q = f(x_q)$ is uniquely determined and can be mapped back to $x_q$ alone. This satisfies the uniqueness criterion.

2. *Proof of Consistency:* A task must be consistent, meaning the underlying rule $f$ applies to all provided example pairs $(x_i, y_i) \in E$ and to the final question-answer pair $(x_q, y_q)$. In our framework, the example set $E$ and the ground-truth answer $y_q$ are not supplied externally but are generated programmatically *after* the function $f$ has been defined and verified. The generation process for any example output $y_i$ is a deterministic execution:

$$y_i := f(x_i), \quad \forall i \in \{1, \ldots, k\} \tag{11}$$

Since each output $y_i$ is the direct result of applying the function $f$ to the input $x_i$, the condition $y_i = f(x_i)$ holds by construction (*ex constructione*). The same applies to the final answer $y_q = f(x_q)$. Therefore, consistency between the rule and all task components is structurally guaranteed by the generation pipeline itself.

3. *Proof of Verifiability:* A task must be verifiable, meaning its logical soundness can be confirmed through a deterministic procedure. The cycle consistency check is precisely this procedure. It is a deterministic, executable program that takes the generated functions $f$ and $g$ as input and tests the property $g(f(x)) = x$. The successful completion of this check serves as a formal, computational certificate of the rule's core logical property—its invertibility via $g$. This programmatic verification replaces subjective human judgment or fallible model-based evaluation with a deterministic and objective confirmation of logical soundness.

Having satisfied all three conditions, any task whose generating functions $(f, g)$ pass the cycle consistency check is guaranteed to be a well-posed problem. □

## A.2. Examples of Invalid Function Mappings

### A.2.1. ONE-TO-MANY MAPPING: VIOLATING UNIQUENESS

A one-to-many mapping violates the Uniqueness criterion because a single input can produce multiple valid outputs, making the correct answer ambiguous.

```python
import random
def f(x):
```

```
    # Non-deterministic: returns a random choice
    return random.choice([x, x+1, x+2])
# Example: f(5) could be 5, 6, or 7
# The "correct" answer is undefined
```

*Listing 1.* A non-deterministic rule that violates uniqueness.

### A.2.2. MANY-TO-ONE MAPPING: CHALLENGING VERIFICATION

A many-to-one mapping is deterministic but loses information, making it impossible to construct a unique inverse function.

```
def f(x):
    # Many inputs map to the same output
    return x % 10

# Example: f(15) = f(25) = f(35) = 5
# Given output 5, we cannot determine if input was 15, 25, or 35
# No unique inverse exists
```

*Listing 2.* A hash-like rule with information loss.

### A.3. Example of Logically Inconsistent Generated Code

Even when an LLM generates syntactically valid Python code, the implementation may contain logical inconsistencies that violate the cycle consistency requirement. Below is a real example from our generation attempts where the forward and inverse functions are incompatible.

```
def rotate_left(s, k):
    n = len(s)
    if n == 0: return s
    k = k % n
    return s[k:] + s[:k]

def transform_grid(grid):
    start = grid.find('<')
    end = grid.find('>')
    if start != -1 and end > start:
        payload = grid[start+1:end]
        k = len(payload) % 3
        v = rotate_left(payload, k)
        return grid[:start] + "{" + str(k) + v + "}" + grid[end+1:]
    return grid

def inverse_transform_grid(grid):
    start = grid.find('{')
    end = grid.find('}')
    if start != -1 and end > start:
        k_val = int(grid[start+1])
        v = grid[start+2:end]
        n = len(v)
        if n == 0: return grid
        shift = k_val % n
        if shift == 0:
            u = v
        else:
            idx = shift - 1
            if idx <= 0:  # Logical error here
                u = v
            else:
                u = v[-idx:] + v[:-idx]  # Incorrect rotation
        return grid[:start] + "<" + u + ">" + grid[end+1:]
```

```
    return grid
```

*Listing 3.* A flawed transformation pair that fails cycle consistency.

**Analysis:** While both functions execute without runtime errors,the inverse function contains a logical flaw in its rotation logic. The cycle consistency check successfully detects this incompatibility, filtering out such tasks automatically.

## B. Methodology and Implementation Details

This section provides a comprehensive overview of the methodological framework and implementation specifics underpinning the A$^2$RBench generation and evaluation pipeline. We detail the experimental setup, including the models employed and their configurations. Furthermore, we present the complete prompt templates that steer the Large Language Models (LLMs) in their designated roles as Authors, Judges, Expanders, and Analysts, offering a transparent view into the mechanisms that drive our automated paradigm.

### B.1. Experimental Setup

Our experimental framework was designed to ensure both diversity in task generation and robustness in evaluation. A diverse set of Large Language Models served in distinct, specialized roles throughout the pipeline.

The generation of the initial seed problems was performed by a cohort of four **Author** models to maximize rule diversity: GPT-5-Mini, Gemini-2.5-Pro, Gemini-2.5-Flash, and O4-mini. For the subsequent evaluation of reasoning capabilities, a broad spectrum of 14 **Solver** models was benchmarked, encompassing the Gemini, GPT, and Qwen series, as enumerated in Table 4. To maintain a consistent and high-quality standard for all auxiliary verification and analysis tasks, a single, powerful model, GPT-5-Mini, was exclusively employed for the roles of **Judge**, **Expander**, and **Analyst**.

All interactions with the LLMs were conducted via API calls, with temperature settings carefully calibrated to the specific nature of each task. We categorized tasks into two primary groups based on their required level of determinism.

- **Tasks Requiring Creativity and Diversity** ($T > 0$)**:** For generative processes where novelty and variation are desirable, a non-zero temperature was employed. This includes the initial rule invention by Author models ('generate rule', $T = 0.5$) and the generation of new input variations by the Expander model ('make request' in task expansion, $T = 0.5$). For code generation ('generate code'), which requires a balance between creativity in implementation and adherence to logical specifications, a slightly lower temperature of $T = 0.2$ was used. The Judge model's preliminary review of rules and code ('judge rule' and 'judge code and output') utilized a minimal temperature of $T = 0.1$ to prevent overly repetitive feedback while maintaining evaluative consistency.

- **Tasks Requiring Determinism and Precision** ($T \approx 0$)**:** For all evaluation and analysis tasks where a single, correct, and reproducible outcome is paramount, a near-zero temperature of $T = 10^{-7}$ was applied. This ensures that the model's output is as deterministic as possible, eliminating randomness from the evaluation process. This setting was used for the Solver models when generating their final answers ('get answer'), for the Judge model's final binary correctness evaluation ('evaluate'), and for the Analyst model's cognitive process classification.

### B.2. Prompt Engineering Framework

The core of our automated benchmark generation pipeline is a sophisticated prompt engineering framework that guides LLMs to perform specialized roles. Each prompt is carefully designed to provide clear instructions, context, and constraints, ensuring high-quality and logically consistent outputs at every stage. This subsection details the specific prompt templates used for the Author, Judge, Expander, and Analyst models.

#### B.2.1. AUTHOR MODEL PROMPTS

The Author model functions as the primary creative engine of the A$^2$RBench pipeline, tasked with conceiving novel, logically sound abstract rules. We decompose the generation process into two distinct interaction turns. In the first turn, the model operates as a "Puzzle Designer," receiving a set of randomly sampled "Inspiration Rules" to prompt the generation of a high-level natural language description of a bijective transformation. In the subsequent turn, the model shifts roles

to that of a "Software Architect," translating the approved description into a deterministic Python implementation. This separation of concerns mitigates the risk of hallucination by grounding the code generation in a pre-validated conceptual framework.

```
System: You are a logical puzzle designer specializing in reversible algorithms. Your task
    ↪  is to invent a novel transformation rule and its corresponding inverse (decoding)
    ↪ rule.

User:
Inspiration Rules (FOR INSPIRATION ONLY):
---
{sampled_rules_str}
---
Your Task:
Invent a new rule for the task type: {task_type}.

CRITICAL DIMENSION REQUIREMENT:
{dimension_instructions}

CRITICAL REQUIREMENT: REVERSIBILITY (BIJECTIVITY)
You are designing a Encoder/Decoder pair.
1.  NO INFORMATION LOSS: The rule must not destroy information.
2.  Forward Rule: Describes how to transform Input A -> Output B.
3.  Inverse Rule: Describes how to verify/reconstruct Input A <- Output B exactly.

Output Format (Strict JSON):
{{
  "reasoning_of_creation": "My reasoning for this reversible {dimensionality} rule...",
  "rule_description": "Clear description of the Forward transformation (Input -> Output)
      ↪ .",
  "inverse_rule_description": "Clear description of the Inverse transformation (Output ->
      ↪ Input)."
}}
```

*Listing 4.* Prompt for Abstract Rule Conception

```
System: You are a meticulous Chief Software Architect. Your goal is to implement a perfect
    ↪ , reversible transformation system in Python. Your output must be self-contained
    ↪ and valid JSON.

User:
Primary Mission: Implement the provided rule as a pair of Python functions.

The Rules You MUST Implement:
1.  Forward Rule: `{rule_description}`
2.  Inverse Rule: `{inverse_rule_description}`

Part 1: The Twin Functions
You must implement TWO functions:
1.  `def transform_grid(grid):` -> Returns the transformed output.
2.  `def inverse_transform_grid(grid):` -> Returns the exact original input*.

Part 2: Robustness
    Handle edge cases gracefully.
    Cycle Consistency: The logic MUST satisfy `inverse_transform_grid(transform_grid(x)) ==
        ↪  x`.

Output Format (Strict JSON with Python string):
{{
  "reasoning": "I have implemented both functions and verified the cycle consistency...",
  "python_code": "import json\n\n# 1. Forward Function\ndef transform_grid(grid):\n..."
}}
```

*Listing 5.* Prompt for Rule Implementation

### B.2.2. JUDGE MODEL PROMPTS (M_JUDGE)

The Judge model acts as a multi-stage quality gate throughout the generation and evaluation process. Its prompts are tailored to three distinct validation tasks: preliminary rule validation, integrated code and puzzle validation, and final answer verification.

After the Author model proposes a rule, the Judge performs an initial logic check to filter out proposals that are ambiguous, non-reversible, or logically flawed. This preemptive check saves computational resources by preventing the generation of code for invalid concepts.

```
System: You are a strict logic judge. Your primary task is to assess if a proposed rule
    ↪ pair is logical, unambiguous, and truly reversible (lossless). Respond in JSON with
    ↪  {'is_valid': boolean, 'reasoning': '...'}.

User:
Review this rule pair.

Forward: `{rule_description}`
Inverse: `{inverse_rule_description}`

Is this rule pair logically sound and verifiably reversible?
```

*Listing 6.* Prompt for Preliminary Rule Validation

Once a rule is approved and code is generated, the Judge conducts a more comprehensive review. This prompt, used in the 'judge code and output' function, instructs the model to verify three key aspects: a) the consistency between the natural language rule and its Python implementation, b) the solvability of the puzzle based on the provided examples, and c) the non-triviality of the task.

```
System: You are a pragmatist logic judge. Your job is to verify that the generated puzzle
    ↪ is solvable and clearly defined.

User:
**Validation Task**
We have a puzzle generated by Python code.

**Rule Definitions:**
- Forward: `{rule_description}`
- Inverse: `{inverse_rule_description}`

**Python Code:**
```python
{python_code}
```

**Generated Puzzle Data:**
```json
{code_output_str}
```

**Your Judgement Criteria:**
1.  **Consistency:** Does the Python code actually implement the Forward and Inverse rules
    ↪  described in natural language?
2.  **Solvability:** Are the provided examples sufficient for a human or AI to deduce the
    ↪ rule? (i.e., The rule isn't "random" or hidden inside the code without external
    ↪ logic).
3.  **Quality:** Is the puzzle non-trivial and interesting?
```

```
**Output Format (Strict JSON):**
```json
{{
  "reasoning": "A concise analysis of rule-to-code consistency, example quality, and
      ↪ overall puzzle non-triviality.",
  "is_valid": <true or false>
}}
```
```

*Listing 7.* Prompt for Integrated Code and Puzzle Validation

During the evaluation phase, the Judge's role simplifies to a direct comparison between the solver model's output and the ground-truth answer. This prompt is designed to be maximally constrained, focusing solely on factual correctness to ensure objective and reproducible scoring.

```
System: You are a meticulous and impartial AI judge. Your task is to determine if a model'
    ↪ s answer is correct based on the provided ground truth. Your output MUST be a
    ↪ single, raw JSON object.

User:
Puzzle Information:
- Rule Description: {rule_description}
- Question: {question_text}

Ground Truth (The official correct answer):
`{ground_truth}`

Model's Answer:
`{model_answer}`

Judging Instructions:
1.  Primary Goal: Determine if the `Model's Answer` is equivalent to the `Ground Truth`.
2.  Equivalence: The answer is correct if it is an exact, character-for-character match.
3.  Reasoning vs. Answer: Your judgment must be based exclusively on the final answer
    ↪ provided, not the model's reasoning process.

Output Format (Strictly adhere to this JSON structure):
{{
    "justification": "<A brief, one-sentence explanation for your decision.>",
    "is_correct": <true or false>
}}
```

*Listing 8.* Prompt for Final Answer Verification

### B.2.3. EXPANDER MODEL PROMPT (M_EXPANDER)

The core responsibility of the M_expander model is to perform systematic data augmentation on each validated seed problem, generating a diverse set of variations designed to probe a solver model's generalization capabilities. To this end, we engineered a structured prompt that instructs the LLM to adopt the persona of a Quality Assurance (QA) Engineer, whose goal is to rigorously test a given rule.

The prompt template, provided in Listing, is designed to guide the LLM's generation process with precision.

```
System: You are a Lead QA Engineer specializing in Abstract Logic Puzzles.
Task: Your job is to generate NEW, DIVERSE input test cases for this rule.
---
Rule Information:
   Rule Description: {rule_description}

Reference Code (Do not modify, use logic for reference):```python
```

```
{python_code}
```
Existing Input History (DO NOT REPEAT THESE):
{input_history}
---
Current Objective (Variation {variation_index}/9):
{variation_guidance}
---
Constraints:
1.  Validity: The new input MUST be valid for the provided code.
2.  Diversity: The new input must be strictly different from any in the History.
3.  Low Entropy Check: If the rule is too restrictive to generate new inputs, you MUST
    ↪ return status "SKIPPED_LOW_ENTROPY".

Output Format (Strict JSON):
{{
  "reasoning": "Explain your strategy for this test case (e.g., testing empty edge case)
    ↪ .",
  "variation_type": "Label for this case (e.g., 'Edge_Case_Empty', 'Adversarial_Pattern')
    ↪ ",
  "new_input": <YOUR_NEW_INPUT_HERE>,
  "status": "CONTINUE"  // or "SKIPPED_LOW_ENTROPY"
}}
```

*Listing 9.* Prompt for Task Expansion

The prompt is composed of four key components to maximize the quality and relevance of the generated data:

**1. Context and Grounding.** To ensure logical fidelity, the prompt grounds the LLM in two ways. First, the **Role-Playing Persona** of a "Lead QA Engineer" conditions the model to shift its operational mode from pure generation to rigorous testing, encouraging it to consider edge cases and adversarial scenarios. Second, and most critically, the **Code as a Logical Oracle** provides the validated Python implementation of the rule. This serves as an unambiguous, deterministic ground truth, compelling the model to generate new inputs that are strictly valid for the given implementation and thereby mitigating the risk of producing fallacious data due to model hallucination.

**2. Dynamic Objective and Phased Strategy.** This is the core mechanism for systematic augmentation. Rather than allowing for unconstrained generation, we dynamically inject specific procedural guidance via the `variation_guidance` placeholder. This guidance is determined by the current `variation_index` (1 through 9) and follows a three-phase strategy:

- **Early Stage (V1-V3):** Focuses on *Standard Variations* to verify baseline comprehension of the rule's core logic.

- **Mid Stage (V4-V6):** Emphasizes *Edge Cases* (e.g., empty structures, singletons, repetitive elements) to test the solver's robustness.

- **Late Stage (V7-V9):** Targets *Complex or Adversarial Cases* to stress-test the solver's reasoning capabilities with inputs of higher complexity or tricky patterns.

This phased approach ensures that the augmented dataset is both structured in its progression of difficulty and diverse in its coverage of the problem space.

**3. State Management and Constraints.** To maintain high data quality, the prompt incorporates two critical constraints. First, by providing the **Input History**, it explicitly instructs the model to generate novel inputs, preventing redundancy. Second, the **Low-Entropy Escape Hatch** serves as a crucial fail-safe mechanism. For rules that are logically restrictive (e.g., "the input must be the exact string 'ABC'"), generating nine unique variations is impossible. This constraint allows the model to gracefully exit by reporting a SKIPPED_LOW_ENTROPY status, which prevents futile retry loops and enhances the overall robustness of the generation pipeline.

**4. Structured Output.** The prompt mandates a strict JSON output format to allow for reliable, automated parsing and validation. The inclusion of `reasoning` and `variation_type` fields provides valuable metadata, enabling detailed downstream analysis of both the generated data and the augmentation model's strategic approach.

In summary, this prompt design channels the generative capabilities of the LLM within a rigorous and deterministic framework. This enables the efficient and systematic generation of high-quality, diverse, and logically-guaranteed test cases for our benchmark.

### B.3. Analyst Model Prompt (`M_analyst`)

The Analyst model serves as a diagnostic instrument designed to look beyond binary accuracy metrics and evaluate the intrinsic quality of the reasoning process. To operationalize the theoretical distinction between superficial pattern matching and genuine algorithmic induction (as discussed in Section 3), this prompt instructs the model to dissect the solver's Chain-of-Thought (CoT).

```
System: You are a meticulous and insightful AI cognitive analyst. Your mission is to
    ↪ dissect a model's thought process to understand not just if it was right, but how
    ↪ it arrived at its conclusion. You must follow the structured classification guide
    ↪ below.

User:
You are given a case file containing a puzzle, the ground truth, and a model's attempt to
    ↪ solve it. Your job is to perform a two-part analysis.

Case File:
- Original Rule Description: {rule_description}
- Ground Truth Answer: {ground_truth}
- Model's Answer: {model_answer}
- Model's Reasoning (Chain of Thought):
{model_cot}

PART 1: Outcome & Reasoning Analysis Guide
PATH A: IF THE MODEL'S ANSWER IS INCORRECT, choose ONE primary cause:
- Abstraction_Failure-Operator_Inference
- Abstraction_Failure-Scope_Condition
- Reasoning_Failure-Procedural_Error
- Format_Or_Collapse-Reasoning_Collapse

PATH B: IF THE MODEL'S ANSWER IS CORRECT, analyze reasoning quality:
- Success-Type_A-Surface_Fitting
- Success-Type_B-Inferior_Rule
- Success-Type_C-Correct_Generalization

PART 2: Reasoning Style Analysis Guide
Independently classify the style of the CoT. Choose ONE:
- Style-Direct_Deduction
- Style-Hypothesis_Testing
- Style-Chaotic_Guessing

Your Response (Strict JSON format):
{{
  "justification": "A brief, one-sentence explanation for your choices.",
  "outcome_category": "CHOSEN_CATEGORY_FROM_GUIDE",
  "reasoning_style": "CHOSEN_STYLE_FROM_GUIDE"
}}
```

*Listing 10.* Prompt for Cognitive Process Analysis

# C. The A$^2$RBench Dataset

This section details the composition and statistical properties of the A$^2$RBench dataset, the primary artifact of our automated generation framework. We first outline its hierarchical structure and scale, followed by a presentation of specific task examples that illustrate the diversity of the benchmark.

## C.1. Dataset Composition and Statistics

A$^2$RBench is constructed as a stratified corpus designed for the rigorous evaluation of algorithmic reasoning across multiple axes of complexity. The final dataset comprises a total of $N_{total} = 1054$ evaluation items, which are organized into two evaluation protocols: a primary set for assessing logical inference ($P_0$), and a diagnostic set for probing symbolic dependency ($P_1$).

The benchmark's architecture is hierarchical. Its foundation is a set of $|T_{seed}| = 72$ unique, code-verified seed tasks (designated V0). Each seed represents a distinct, formally validated abstract rule. Through the systematic augmentation pipeline outlined in Section 4, these seeds are programmatically expanded into $|T_{aug}| = 631$ augmented instances (V1–V9). This process yields a comprehensive base library for the primary evaluation protocol, totaling $|P_0| = 703$ unique reasoning problems. To facilitate the diagnostic analysis of symbolic dependency, the $P_1$ protocol was created by applying isomorphic symbol perturbations to all tasks within the Symbolic domain of $P_0$. This results in an additional $|P_1| = 351$ evaluation items, establishing a one-to-one task for each symbolic task.

A defining characteristic of A$^2$RBench is its principled stratification across both logical dimensions and semantic domains. Tasks are categorized into two primary domains: Symbolic Rules, which operate on the structure and arrangement of abstract tokens, and Semantic Rules, which require the application of external world knowledge (e.g., lexical sequences or arithmetic properties). As detailed in Table 2, this domain-level division is cross-balanced with a nearly uniform distribution across three spatial dimensions: 1D (Sequence), 2D (Grid), and 3D (Voxel). This balanced design, with 237 tasks in 1D, 237 in 2D, and 229 in 3D, is critical for isolating model capabilities from data distribution biases, thereby ensuring that observed performance variations, such as the generative bottleneck discussed in Section 5, can be confidently attributed to the models themselves.

*Table 2.* **Distribution of Reasoning Tasks in the $P_0$ Protocol.** The dataset maintains a balanced stratification across both rule types (Symbolic vs. Semantic) and spatial dimensions (1D, 2D, 3D), minimizing confounding variables in the evaluation of reasoning stability.

| RULE TYPE | 1D (SEQ) | 2D (GRID) | 3D (VOXEL) | TOTAL |
|---|---|---|---|---|
| SYMBOLIC RULE | 117 | 120 | 114 | **351** |
| SEMANTIC RULE | 120 | 117 | 115 | **352** |
| **TOTAL** | **237** | **237** | **229** | **703** |

## C.2. Exemplary Tasks from the A$^2$RBench Benchmark

To provide a concrete illustration of the tasks generated by our framework, this section presents six representative examples authored by the `Author_O4_mini` model. These examples are selected to showcase the diversity of the benchmark, spanning both symbolic and semantic rule types across one, two, and three dimensions. Each task is presented in its raw JSON format, encapsulating the set of examples required for rule induction, alongside the specific question and its ground-truth answer, all programmatically verified for logical consistency via the Cycle Consistency constraint.

### C.2.1. SYMBOLIC TRANSFORMATION RULES

Symbolic tasks operate on the structure and arrangement of input symbols, without reference to their external meaning.

This one-dimensional task involves a structural rearrangement rule that interleaves the two halves of an input sequence.

```
{
  "examples": [
    {"input": [], "output": []},
    {"input": ["x"], "output": ["x"]},
    {"input": ["a", "b", "c", "d"], "output": ["c", "a", "d", "b"]},
```

```
        {"input": ["a", "b", "c", "d", "e"], "output": ["d", "a", "e", "b", "c"]}
    ],
    "question_plaintext": ["p", "y", "t", "h", "o", "n", "3"],
    "answer_ciphertext": ["o", "p", "n", "y", "3", "t", "h"]
}
```

*Listing 11.* A 1D symbolic task involving sequence interleaving.

The two-dimensional symbolic task applies a local pairwise swap within every non-overlapping 2×2 block: the top-right and bottom-left elements exchange positions, while the top-left and bottom-right remain fixed. Rows or columns that do not form complete 2×2 blocks are left unchanged.

```
{
  "examples": [
    {"input": [], "output": []},
    {"input": [[1]], "output": [[1]]},
    {"input": [[1, 2], [3, 4]], "output": [[1, 3], [2, 4]]},
    {"input": [[1, 2, 3], [4, 5, 6]], "output": [[1, 4, 3], [2, 5, 6]]},
    {"input": [[1, 2, 3], [4, 5, 6], [7, 8, 9]],
     "output": [[1, 4, 3], [2, 5, 6], [7, 8, 9]]}
  ],
  "question_plaintext": [
    [1, 2, 3, 4],
    [5, 6, 7, 8],
    [9, 10, 11, 12],
    [13, 14, 15, 16],
    [17, 18, 19, 20]
  ],
  "answer_ciphertext": [
    [1, 5, 3, 7],
    [2, 6, 4, 8],
    [9, 13, 11, 15],
    [10, 14, 12, 16],
    [17, 18, 19, 20]
  ]
}
```

*Listing 12.* A 2D symbolic task based on block-wise diagonal swaps.

### C.2.2. SEMANTIC TRANSFORMATION RULES

Semantic tasks require the application of external knowledge or meaning associated with the symbols to derive the correct transformation.

This semantic task requires external knowledge, mapping each letter to a chemical element symbol where the letter's alphabetical position corresponds to the element's atomic number.

```
{
  "examples": [
    {"input": ["CAT"], "output": ["LiHCa"]},
    {"input": ["HELLO"], "output": ["OBMgMgP"]}
  ],
  "question_plaintext": ["BACK", "CAGE"],
  "answer_ciphertext": ["HeHLiNa", "LiHNB"]
}
```

*Listing 13.* A 1D semantic task mapping letters to chemical elements via atomic number.

The two-dimensional semantic rule is an involution that combines a 180-degree spatial rotation of the grid with a mirrored character substitution cipher applied to letters, digits, and paired brackets.

```
{
  "examples": [
    {"input": ["AB", "CD"], "output": ["WX", "YZ"]},
    {"input": ["A"], "output": ["Z"]}
  ],
  "question_plaintext": ["ABC123XYZ0", "12345-6789", "()[]{}<>!?"],
  "answer_ciphertext": ["?!<>{}[]()", "0123-45678", "9ABC678XYZ"]
}
```

*Listing 14.* A 2D semantic task integrating 180-degree rotation with a mirror cipher.

The three-dimensional semantic task integrates a 90-degree clockwise rotation about the vertical axis with a subsequent application of the Atbash cipher to all alphabetic characters within the cube.

```
{
  "examples": [
    {"input": [[["M"]]], "output": [[["N"]]]},
    {"input": [[["A", "b"], ["C", "1"]], [["x", "!"], ["Z", "z"]]], "output": [[["c", "Z
        ↪ "], ["A", "X"]], [["!", "y"], ["a", "1"]]]}
  ],
  "question_plaintext": [[["A", "b"], ["C", "1"]], [["x", "!"], ["Z", "z"]]],
  "answer_ciphertext": [[["c", "Z"], ["A", "X"]], [["!", "y"], ["a", "1"]]]
}
```

*Listing 15.* A 3D semantic task combining a Y-axis rotation with the Atbash cipher.

### C.3. Additional Task Examples

This section presents two supplementary examples to further illustrate the diversity and complexity spectrum of the A$^2$RBench benchmark. Both tasks are authored by the `Author_O4_mini` model and represent problems within the two-dimensional space. The first demonstrates a composite structural transformation through sequential row and column operations, while the second exemplifies the integration of cryptographic primitives with spatial manipulation. As with all tasks in our benchmark, these problems have been formally verified through programmatic cycle consistency checks.

This two-dimensional symbolic task applies a composition of circular shifts: first, each row $i$ is rotated right by $(i \bmod N)$ positions; then, each column $j$ in the resulting grid is rotated downward by $(j \bmod M)$ positions. The transformation requires tracking both row-wise and column-wise index-dependent operations.

```
{
  "examples": [
    {"input": [], "output": []},
    {"input": [["X"]], "output": [["X"]]},
    {"input": [["a", "b", "c"], ["d", "e", "f"], ["g", "h", "i"]],
     "output": [["a", "i", "e"], ["f", "b", "g"], ["h", "d", "c"]]}
  ],
  "question_plaintext": [["t", "h", "i", "s"],
                         ["i", "s", "a", "t"],
                         ["e", "s", "t", "g"]],
  "answer_ciphertext": [["t", "g", "s", "s"],
                        ["t", "h", "e", "a"],
                        ["t", "i", "i", "s"]]
}
```

*Listing 16.* A 2D symbolic task combining row and column circular shifts.

This two-dimensional semantic task integrates cryptographic transformation with spatial manipulation: each character undergoes an Atbash cipher for letters (A↔Z, a↔z) and digit complement for numbers (0↔9), followed by a 180-degree rotation of the entire grid (reversing both row order and character order within each row). The transformation is involutive.

```
{
  "examples": [
    {"input": ["ABC", "xyz", "123"],
     "output": ["678", "abc", "XYZ"]},
    {"input": ["Hello, World!", "Atbash 123?"],
     "output": ["?678 shzygZ", "!woilD ,loovS"]}
  ],
  "question_plaintext": ["FooBar", "Baz123!", ""],
  "answer_ciphertext": ["", "!678azY", "izYllU"]
}
```

*Listing 17.* A 2D semantic task combining Atbash cipher with 180-degree rotation.

### C.4. Cost Analysis

A critical advantage of our automated generation paradigm is its economic efficiency. Traditional reasoning benchmarks face a fundamental trade-off between scale and cost: manually designed benchmarks like ARC require substantial time investment from domain experts, while large-scale datasets such as GSM8K necessitate professional annotators at rates of tens of dollars per hour. In contrast, our pipeline leverages LLM-based generation with formal verification, achieving scalability at a fraction of the cost.

Table 3 presents a comparative cost analysis across three representative benchmarks. Since ARC and GSM8K do not publicly disclose their annotation costs, we estimate based on typical expert annotation rates ($50/hour) and task complexity. ARC's visual-spatial puzzles require extensive design time (30-60 minutes per task), yielding an estimated cost of $25-$50 per problem. GSM8K's mathematical word problems, while less complex, still demand professional annotation at approximately $8-$12 per task. In stark contrast, our automated pipeline achieves dramatically lower costs through systematic reuse of validated rules.

Seed generation, which involves multi-stage LLM verification (rule design, code implementation, and cycle consistency checks), costs $0.19 per task. Once a seed is validated, however, the expansion phase becomes remarkably efficient: generating 631 augmented variations costs merely $0.005 per task—a $38\times$ reduction. Averaged across the entire dataset of 1,054 tasks, the cost is $0.016 per problem. This economic efficiency, combined with formal guarantees of logical soundness, establishes a scalable paradigm for benchmark construction that maintains cognitive rigor without the prohibitive costs of manual expert curation.

*Table 3.* Cost comparison across reasoning benchmarks. ARC and GSM8K costs are estimated based on typical expert annotation rates ($50/hour) and task design time. $A^2$RBench costs are actual API expenses.

| Benchmark | Dataset Size | Est. Cost per Task | Total Est. Cost |
|---|---|---|---|
| ARC | 1,000 | $25–$50 | $25,000–$50,000 |
| GSM8K | 8,500 | $8–$12 | $68,000–$102,000 |
| $A^2$RBench (Ours) | 1,054 | $0.016 | $16.86 |
| - Seed (V0) | 72 | $0.19 | $13.68 |
| - Augmented (V1-V9) | 631 | $0.005 | $3.16 |

## D. Detailed Experimental Results and Analysis

This section provides a detailed repository of the quantitative results from our extensive model evaluations. To facilitate clear interpretation and direct linkage to the core findings presented in the main paper, the data is organized thematically. We begin with the global performance leaderboard, followed by granular analyses that deconstruct model performance across logical dimensions, the effects of task augmentation, and the underlying complexity of the generated rules.

## D.1. Global Performance

Table 4 presents the comprehensive performance metrics for all 14 evaluated models across the entirety of the A$^2$RBench dataset ($N = 1054$). This table serves as the primary leaderboard, summarizing overall accuracy, performance on symbolic versus semantic tasks, and the generalization gap between seed (V0) and augmented (V1-V9) tasks. Crucially, it quantifies the Symbolic Dependency Gap ($\Delta_S$), which measures the performance degradation on symbol-remapped tasks and serves as a key indicator of models' over-reliance on familiar tokens.

*Table 4.* **Global Performance Leaderboard.** Detailed accuracy metrics across all 1054 tasks. **Total Acc**: Overall accuracy. **Sym Acc**: Accuracy on all symbolic tasks. **Sem Acc**: Accuracy on semantic tasks. **Gap ($\Delta_S$)**: The performance drop on symbolic tasks when symbols are remapped ('Original Acc - Mapped Acc'), serving as a proxy for "Illusion of Understanding".

| MODEL | TOTAL ACC | SYM ACC | SEM ACC | SEED ACC (V0) | AUG. ACC (V1-V9) | GAP ($\Delta_S$) | COLLAPSE RATE |
|---|---|---|---|---|---|---|---|
| **GEMINI3-PRO** | **40.9%** | **37.0%** | 48.6% | **39.8%** | **41.0%** | 4.6% | 39.7% |
| GPT-5 | 39.0% | 32.5% | **52.0%** | 38.9% | 39.0% | 17.7% | 23.2% |
| GPT-5-MINI | 36.9% | 31.5% | 47.7% | 34.3% | 37.2% | 17.4% | 8.0% |
| O4-MINI | 33.7% | 28.3% | 44.3% | 34.3% | 33.6% | 16.2% | 16.0% |
| CLAUDE-SONNET-4.5 | 30.6% | 33.3% | 25.3% | 23.1% | 31.5% | 1.7% | 8.2% |
| GPT-5.2 | 28.3% | 30.5% | 23.9% | 21.3% | 29.1% | 4.0% | 1.1% |
| GEMINI-2.5-FLASH | 27.7% | 26.4% | 30.4% | 25.0% | 28.0% | 6.0% | 24.3% |
| GLM-4.6 | 21.0% | 23.1% | 16.8% | 13.9% | 21.8% | 3.4% | 12.7% |
| DEEPSEEK-V3.2 | 20.8% | 20.8% | 20.7% | 17.6% | 21.1% | 5.1% | 3.0% |
| QWEN3-32B | 19.3% | 22.5% | 12.8% | 6.5% | 20.7% | 1.1% | 9.7% |
| GPT4O-MINI | 16.1% | 19.5% | 9.4% | 8.3% | 17.0% | 8.8% | 10.0% |
| QWEN3-14B | 14.2% | 17.0% | 8.8% | 5.6% | 15.2% | -0.3% | 9.7% |
| GEMINI3-FLASH | 13.1% | 13.0% | 13.4% | 11.1% | 13.3% | 5.4% | 84.7% |
| QWEN3-8B | 0.0% | 0.0% | 0.0% | 0.0% | 0.0% | 0.0% | 100.0% |

**GPT Family Seed-Subset Analysis.** We further evaluated GPT-family models on the 108-task seed subset. The GPT family is non-monotonic: gpt-5 achieves the highest accuracy and uses the largest average total token budget, while gpt-5.2 and gpt-5.4 use nearly the same average total tokens but differ substantially in accuracy. This suggests that the difference is better explained by reasoning budget and solution strategy than by a simple monotonic capability ordering.

*Table 5.* GPT-family results on the 108-task seed subset.

| Model | Accuracy | Avg. Total Tokens |
|---|---|---|
| gpt-5 | 37.96% | 3175.1 |
| gpt-5-mini | 33.33% | 2132.1 |
| gpt-5.4 | 23.15% | 989.3 |
| gpt-5.2 | 17.59% | 992.9 |
| gpt-5.4-mini | 11.11% | 934.9 |

## D.2. Human Agreement Validation

We validate the Analyst LLM on stratified human-annotated subsets. For the success taxonomy, we sample 90 successful responses, with 30 examples each from True Generalization, Inferior Rule, and Surface Fitting. For the error taxonomy, we sample 90 incorrect responses, with 30 examples each from Cognitive Collapse, Abstraction Failure, and Execution Error. A computer science undergraduate independently blind-annotates these samples and we compare the labels with GPT-5-mini. These agreement rates provide encouraging support for the relevant judgment stages and suggest good practical reliability, but we do not claim that the LLM judge is noise-free or fully equivalent to human annotation.

*Table 6.* Human agreement validation for Analyst LLM taxonomies.

| Taxonomy | Agreement | Cohen's $\kappa$ | Macro-F1 |
|---|---|---|---|
| Success taxonomy | 76/90 (84.44%) | 0.7667 | 0.8449 |
| Error taxonomy | 75/90 (83.33%) | 0.7500 | 0.8336 |

### D.3. Fine-Tuning with $A^2$RBench Supervision

We conducted a LoRA fine-tuning experiment using Qwen3-8B on $A^2$RBench supervision, where each instance contains the problem, example input-output pairs, and the answer. To avoid leakage from near-duplicate variants, we split by rule family, yielding 673 training examples and 30 validation examples. We use LoRA with $r = 16$, $\alpha = 32$, dropout 0.05, learning rate $2 \times 10^{-4}$, a cosine scheduler, and effective batch size 8; generation uses max sequence length 3072, max new tokens 512, temperature 0, and thinking mode disabled. The best checkpoint is epoch 2. The gains are not uniform across reasoning skills, but suggest improved answer convergence, output alignment, and discrete rule discrimination.

Table 7. Fine-tuning results with $A^2$RBench supervision.

| Metric | Result |
|---|---|
| Best checkpoint | epoch 2 |
| MMLU-Pro General | 16.35% $\rightarrow$ 34.13% |
| BBH Reasoning | 9.15% $\rightarrow$ 13.24% |
| BBH formal_fallacies | 2.8% $\rightarrow$ 59.2% |
| BBH navigate | 10.8% $\rightarrow$ 59.6% |
| BBH web_of_lies | 21.2% $\rightarrow$ 41.6% |
| BBH boolean_expressions | 64.0% $\rightarrow$ 82.8% |
| BBH JSON alignment | 29.04% $\rightarrow$ 100.00% |
| MMLU JSON alignment | 13.54% $\rightarrow$ 99.95% |

### D.4. Dimensionality Bottleneck

Our results reveal a consistent performance dip on 2D tasks, creating a 1D > 3D > 2D performance hierarchy (Figure 3). This 'V-shaped' trough implies that 3D tasks are inadvertently easier than 2D ones. Our analysis confirms this is not due to the intrinsic nature of the dimensions, but rather an artifact of the interaction between the author models' generative capabilities and the solvers' weaknesses.

Table 8 breaks down solver performance by the author of each task. The data indicates that the performance drop in 2D is primarily driven by tasks from a single author: O4-mini. On 2D tasks authored by the O4-mini author model, the accuracy of top models like Gemini3-Pro and GPT-5 plummets to just **22.2%** and **23.3%**, respectively. While other authors (e.g., Gemini-Flash) also exhibit a trend where 3D tasks are solved more accurately than 2D tasks, this gap is far less pronounced compared to the drastic collapse observed with O4-mini.

This phenomenon exposes a critical "Generative Capability Ceiling" in current LLMs. As shown in our AST analysis (Table 9), O4-mini's 2D tasks exhibit exceptionally high logical complexity, with the highest 'Nested If Depth' (2.33) and 'Return Complexity' (15.67). However, when operating in 3D, these complexity metrics drop significantly.

We infer that this represents a cognitive trade-off: the author model cannot simultaneously maximize spatial complexity and logical complexity. Managing the additional spatial dimension (z-axis) in 3D consumes the model's reasoning ability, forcing it to simplify the underlying conditional logic to maintain validity. O4-mini, being a stronger model, pushes the logic complexity to the limit in 2D, making its subsequent "simplification" in 3D—and the resulting performance gap—much more significant than weaker models like Gemini-Flash, which generate simpler code across all dimensions.

### D.5. Augmentation Paradox

The "Augmentation Paradox," where certain algorithmically complex inputs paradoxically simplify the reasoning task, is quantitatively detailed in Table 10. This table correlates the information complexity of each problem variation (V0-V9), measured by compression ratio, with empirical failure metrics from model responses. The data starkly reveals that variations with the highest input complexity (e.g., V4, an edge case) simultaneously exhibit the lowest failure entropy and the highest solver accuracy, supporting our conclusion that structured, adversarial inputs can inadvertently provide strong cues that reduce rule ambiguity.

*Table 8.* Solver Performance Across Dimensions (1D, 2D, 3D) for Each Author Model.

| SOLVER MODEL | GPT-5-MINI | | | GEMINI2.5-FLASH | | | GEMINI2.5-PRO | | | O4-MINI | | |
|---|---|---|---|---|---|---|---|---|---|---|---|---|
| | 1D | 2D | 3D | 1D | 2D | 3D | 1D | 2D | 3D | 1D | 2D | 3D |
| GEMINI3-PRO | 63.3% | 43.3% | 36.0% | 27.8% | 42.2% | 42.2% | 42.9% | 46.0% | 28.7% | 58.9% | 22.2% | 35.7% |
| GPT-5 | 52.2% | 36.7% | 44.9% | 38.9% | 35.6% | 41.1% | 36.9% | 42.5% | 27.5% | 50.0% | 23.3% | 36.9% |
| GPT-5-MINI | 47.8% | 33.3% | 39.3% | 46.7% | 27.8% | 43.3% | 39.3% | 29.9% | 27.5% | 50.0% | 23.3% | 33.3% |
| O4-MINI | 52.2% | 25.6% | 28.1% | 33.3% | 23.3% | 38.9% | 41.7% | 28.7% | 28.7% | 51.1% | 20.0% | 32.1% |
| CLAUDE-SONNET 4.5 | 51.1% | 27.8% | 24.7% | 31.1% | 24.4% | 15.6% | 46.4% | 26.4% | 18.8% | 54.4% | 21.1% | 25.0% |
| GPT-5-2 | 40.0% | 30.0% | 28.1% | 33.3% | 18.9% | 20.0% | 42.9% | 26.4% | 17.5% | 41.1% | 15.6% | 25.0% |
| GEMINI2.5-FLASH | 53.3% | 25.6% | 11.2% | 23.3% | 22.2% | 28.9% | 33.3% | 25.3% | 26.2% | 46.7% | 13.3% | 22.6% |
| GLM-4.6 | 26.7% | 23.3% | 21.3% | 22.2% | 16.7% | 10.0% | 32.1% | 16.1% | 18.8% | 33.3% | 13.3% | 17.9% |
| DEEPSEEK-V3-2 | 32.2% | 23.3% | 19.1% | 18.9% | 22.2% | 14.4% | 29.8% | 27.6% | 17.5% | 24.4% | 13.3% | 6.0% |
| QWEN3-32B | 26.7% | 18.9% | 16.9% | 24.4% | 12.2% | 11.1% | 31.0% | 11.5% | 16.2% | 22.2% | 15.6% | 25.0% |
| GPT4O-MINI | 22.2% | 13.3% | 19.1% | 15.6% | 7.8% | 10.0% | 22.6% | 12.6% | 15.0% | 17.8% | 13.3% | 25.0% |
| QWEN3-14B | 17.8% | 15.6% | 14.6% | 17.8% | 15.6% | 6.7% | 25.0% | 3.4% | 11.2% | 15.6% | 10.0% | 17.9% |
| GEMINI3-FLASH | 7.8% | 16.7% | 11.2% | 13.3% | 4.4% | 7.8% | 23.8% | 4.6% | 10.0% | 20.0% | 13.3% | 25.0% |
| QWEN3-8B | 0.0% | 0.0% | 0.0% | 0.0% | 0.0% | 0.0% | 0.0% | 0.0% | 0.0% | 0.0% | 0.0% | 0.0% |

*Table 9.* **AST-based Complexity Metrics of Generated Rules by Author Model.** This table details the average complexity of the Python code generated for 1D, 2D, and 3D tasks. To highlight the generative bottleneck, values in **bold** indicate the higher complexity score when comparing 2D and 3D tasks for each metric. The data reveals a nuanced picture: while some models generate 3D tasks with deeper loop structures (e.g., GPT-5-Mini), O4-mini produces 2D tasks with significantly more complex conditional logic (Nested If Depth) and return statements (Return Complexity). This explains why solver models struggle specifically with O4-mini's 2D tasks.
*Metrics*: **Max Loop Depth**: Maximum level of nested loops. **Total Ifs**: Total number of 'if'/'elif' statements. **Nested If Depth**: Maximum depth of nested conditionals. **Cond. Complexity**: Max Cyclomatic complexity. **Mutability Score**: Frequency of in-place modifications. **Return Complexity**: Cyclomatic complexity of the return statement.

| Author Model | Dim. | Max Loop Depth | Total Ifs | Nested If Depth | Cond. Complexity | Mutability Score | Return Complexity |
|---|---|---|---|---|---|---|---|
| GPT-5-Mini | 1D | 1.00 | 3.83 | 1.17 | 11.50 | 0.50 | 11.17 |
| | 2D | 1.00 | 3.50 | 1.00 | 9.00 | 0.00 | 2.00 |
| | 3D | **2.50** | **8.00** | **4.25** | 9.00 | **0.75** | 2.00 |
| Gemini-2.5-Flash | 1D | 0.50 | 2.67 | 1.17 | 8.33 | 0.00 | 6.83 |
| | 2D | 1.25 | 3.00 | 1.00 | 8.75 | 0.00 | 2.00 |
| | 3D | **2.33** | **3.33** | **1.33** | **11.33** | 0.00 | 2.00 |
| Gemini-2.5-Pro | 1D | 0.20 | 2.40 | 1.00 | 11.80 | 0.00 | 11.80 |
| | 2D | 0.00 | 1.50 | 1.00 | 6.00 | 0.00 | **9.75** |
| | 3D | 0.00 | **1.60** | 1.00 | 6.00 | 0.00 | 7.80 |
| O4-mini | 1D | 0.67 | 1.83 | 1.17 | 8.00 | 0.00 | 7.17 |
| | 2D | 1.67 | 3.00 | **2.33** | **7.67** | 1.33 | **15.67** |
| | 3D | **2.80** | 3.00 | 1.40 | 7.20 | **1.40** | 2.00 |

## D.6. Human vs. Model

To ground our findings in the context of human-level intelligence, we conducted a comparative evaluation on a representative subset of 108 seed tasks. Table 11 directly compares the performance of our top-performing model, Gemini3-Pro, against human participants. The results highlight a significant performance delta across all categories—overall, by rule type, and by dimensionality—underscoring the substantial gap that currently exists between state-of-the-art LLMs and human abstract reasoning capabilities.

The human study includes three participant groups with different technical backgrounds: five CS PhD participants, five CS undergraduate participants, and five non-CS undergraduate participants. All participants completed the same 108 seed tasks under no time limit, so the comparison should be interpreted as a stratified human baseline rather than a claim about a single universal human level.

*Table 10.* **Quantitative Analysis of Problem Variation Complexity and Empirical Failure Modes.** This table correlates the intrinsic information complexity of each problem variation (V0-V9) with the diversity and predictability of failure modes exhibited by all evaluated models. The data reveals a significant negative correlation between information complexity (compression ratio) and failure entropy, providing quantitative evidence for the "Augmentation Paradox" proposed in the paper. The peak of this effect is observed in variation **V4**, which has the highest information complexity and the lowest failure entropy.

| PROBLEM VARIATION | INFORMATION COMPLEXITY COMPRESSION RATIO[1] | EMPIRICAL FAILURE ANALYSIS UNIQUE ERRORS[2] | FAILURE ENTROPY (BITS)[3] | AVG. EDIT DISTANCE[4] |
|---|---|---|---|---|
| V0 (SEED) | 1.156 | 10.264 | 2.825 | 56.221 |
| V1 (STANDARD) | 1.181 | 10.380 | 2.850 | 58.102 |
| V2 (STANDARD) | 1.179 | 10.211 | 2.811 | 57.279 |
| V3 (STANDARD) | 1.182 | 10.620 | 2.882 | 57.036 |
| V4 (EDGE CASE) | **4.286** | **4.181** | **1.532** | 68.782 |
| V5 (EDGE CASE) | 3.346 | 5.139 | 1.751 | 66.292 |
| V6 (EDGE CASE) | 2.781 | 5.746 | 1.890 | 63.807 |
| V7 (ADVERSARIAL) | 0.769 | 11.676 | 2.935 | 54.012 |
| V8 (ADVERSARIAL) | 0.641 | 11.118 | 2.867 | 57.867 |
| V9 (ADVERSARIAL) | 0.565 | 10.433 | 2.688 | 60.455 |

NOTES:
[1] COMPRESSION RATIO: THE RATIO OF THE GZIP-COMPRESSED SIZE TO THE ORIGINAL SIZE OF THE INPUT STRING. A HIGHER RATIO INDICATES LOWER PATTERN REGULARITY AND THUS HIGHER INFORMATION COMPLEXITY.
[2] UNIQUE ERRORS: THE AVERAGE COUNT OF DISTINCT INCORRECT ANSWERS PRODUCED BY ALL MODELS FOR A GIVEN VARIATION, MEASURING THE DIVERGENCE OF FAILURE MODES.
[3] FAILURE ENTROPY: THE SHANNON ENTROPY CALCULATED FROM THE DISTRIBUTION OF INCORRECT ANSWERS, QUANTIFYING THE UNCERTAINTY OF A MODEL'S FAILURE MODE. LOWER ENTROPY SIGNIFIES MORE CONCENTRATED AND PREDICTABLE FAILURES.
[4] AVG. EDIT DISTANCE: THE AVERAGE STRING EDIT DISTANCE BETWEEN A MODEL'S INCORRECT OUTPUT AND THE GROUND-TRUTH ANSWER.

*Table 11.* Performance Comparison of Human vs. Gemini3-Pro on a representative subset of 108 tasks from the $A^2$RBench benchmark. This test set comprises all 36 semantic seed tasks, along with the 36 symbolic seed tasks presented in both their original and symbol-remapped forms. The results highlight a significant performance gap.

| PERFORMANCE METRIC | | GEMINI 3 PRO | HUMAN |
|---|---|---|---|
| OVERALL PERFORMANCE | | 39.81% | **68.52%** |
| BY RULE TYPE | SYMBOLIC | 34.72% | **56.94%** |
| | SEMANTIC | 50.00% | **91.67%** |
| BY DIMENSIONALITY | 1D | 50.00% | **72.22%** |
| | 2D | 33.33% | **63.89%** |
| | 3D | 36.11% | **69.44%** |

*Table 12.* Human participant groups and within-group agreement. Each group contains five participants on the 108-task subset.

| Group | #Tasks | Overall Acc. | 1D | 2D | 3D |
|---|---|---|---|---|---|
| CS PhD | 540 | 370/540 (68.52%) | 170/180 (94.44%) | 159/180 (88.33%) | 41/180 (22.78%) |
| CS Undergraduate | 540 | 261/540 (48.33%) | 160/180 (88.89%) | 83/180 (46.11%) | 18/180 (10.00%) |
| Non-CS Undergraduate | 540 | 136/540 (25.19%) | 111/180 (61.67%) | 15/180 (8.33%) | 10/180 (5.56%) |

