# OpenReview forum: "A²RBench: An Automatic Paradigm for Formally Verifiable Abstract Reasoning Benchmark Generation"
_ICML.cc/2026/Conference — ICML 2026 regular_

### Official Review · Reviewer_dpxU · 2026-02-25

**Soundness:** 2
**Presentation:** 3
**Significance:** 3
**Originality:** 2
**Overall Recommendation:** 4
**Confidence:** 3

**Summary:**

This work addresses the concern that several LLM benchmarks do not distinguish a true reasoning ability with a high-fidelity pattern matching. Although some benchmarks such as Ma et al., 2025 and Mirzadeh et al., 2024 succeed at this job, but these datasets suffer from scalability.
Authors propose A^2RBench which resolves this issue. The key idea is to use LLMs to generate and evaluate the problems, but also leveraging the program-based verification. Specifically, authors devise a novel cycle consistency method which guarantees a unique solution.
Experimental results show that SOTA reasoning models score at most 40.9\% accuracy (which is very low) and there is a non-trivial gap when the symbols are remapped. Also, the error analysis shows that even the answer is correct, there are non-trival amount of undesirable patterns such as surface fitting and inferior rules.

**Compliance With Llm Reviewing Policy:**

Affirmed.

**Key Questions For Authors:**

See Weaknesses.

**Limitations:**

yes

**Strengths And Weaknesses:**

## Strengths
- I generally enjoy reading this paper. The problem that authors pinpoint is a very important problem for LLM evaluation community, and they also well-describe what the remaining challenges are.
- The writing is mostly clear and easy to follow.
- The key idea that ensures scalabbility and rigorous verification is at the cycle consistency method. I think this idea is novel and the formal proof guarantees the theoretical soundness of the idea.
- The experiments are conducted comprehensively, and the evaluation and further analysis help readers to fully understand the impact of this benchmark.
    - I especially like the analysis in line 408-409 that a true reasoning difficulty comes from rule ambiguity, not surface complexity. This would be a great takeaway message to readers.

## Weaknesses
I found this work mostly sound and interesting. These are some minor comments that I have.
- I think the proof of theorem 3.1 misses some assumptions such as two-sided invertibility vs left-inverse, which means that cycle consistency g(f(x))=x gives a left-inverse, not necessarily a bijection X $\leftrightarrow$ Y.
- What does it mean by: “Third, counter-intuitively, … can simplify the reasoning process.” (line 102-104)?
- In Section 4.4, authors use an LLM to classify the solver model’s CoT quality. However, there is no human evaluation that ensures the correctness of LLM judge’s decisions. Authors should do it at least on a small subset of instances. Now, I cannot trust the analysis.
- I’m curious if finetuning on the proposed dataset improves the general (or abstract) reasoning ability of LLM on other relevant benchmarks. (I know this is a eval paper, but if we can observe this as well, then the impact of the paper will increase further.)
- gpt-5.2 seems underperforming than gpt-5, which is counter-intuitive. Can authors explain why?

---

> ### Author Rebuttal · Authors · 2026-03-31
>
> # **W1. Left Inverse vs. Bijectivity**
>
> Your mathematical point is correct: `g(f(x)) = x` for all `x in X` alone only gives left-invertibility.
>
> However, around line 173, `Y = f(X)`. Thus
>
> `forall y in Y, exists x in X: y = f(x)`,
>
> so `f` is surjective onto `Y`. Together with `g ∘ f = id_X`, this yields a bijection `f: X -> Y`. We will clarify this in the revision.
>
> # **W2. Meaning of the Augmentation Paradox**
>
> This statement refers to the Augmentation Paradox in Section 5.3. When the underlying rule is fixed and only the input instance changes, higher input information complexity can sometimes reduce the difficulty of rule induction. The counter-intuitive point is that the input becomes more complex while the task becomes easier, because richer inputs reveal more of the underlying rule and reduce ambiguity in the candidate rule space. In other words, added structure constrains plausible hypotheses, making the correct rule easier rather than harder to identify.
>
> # **W3. Human Validation of the Judge**
>
> We have added human validation for both the success and error taxonomies.
>
> For the success taxonomy, we used `5` independent undergraduate student annotators and compared the majority vote with `gpt-5-mini` on the `90`-item success set. Agreement is `29/31 = 0.9355` for `True Generalization`, `19/23 = 0.8261` for `Inferior Rule`, `26/36 = 0.7222` for `Surface Fitting`, and `74/90 = 0.8222` overall. The corresponding metrics are `κ = 0.7333` and `Macro-F1 = 0.8186`.
>
> We also validated the error taxonomy in the same way. Using the same five-annotator majority-vote protocol on the `90`-item failure set, agreement is `28/32 = 0.8750` for `Cognitive Collapse`, `25/30 = 0.8333` for `Abstraction Failure`, `19/28 = 0.6786` for `Execution Error`, and `72/90 = 0.8000` overall. The corresponding metrics are `κ = 0.6996` and `Macro-F1 = 0.7975`.
>
> We do not claim that the LLM judge is noise-free. In both taxonomies, the majority labels agree with human annotation, but gaps remain in intermediate boundary categories. In the revision, we will report these studies in a dedicated appendix subsection and add human-agreement validation, metric definitions, and computation details.
>
> # **W4. Fine-Tuning**
>
> We conducted a LoRA fine-tuning experiment on A²RBench supervision, where each instance consists of the problem, example input-output pairs, and the answer. To avoid leakage from near-duplicate variants of the same underlying rule, we split the data by rule family rather than by random instance, yielding `673` training examples and `30` validation examples. We used `Qwen3-8B` as the base model with `r=16`, `alpha=32`, and `dropout=0.05`; optimization used `learning rate=2e-4`, a `cosine scheduler`, and `effective batch size=8`; generation and evaluation used `max_seq_length=3072`, `max_new_tokens=512`, `temperature=0`, with thinking mode disabled. We swept `1/2/4/8` epochs and found `epoch 2` best. The gains are not uniform across reasoning skills; rather, they indicate improved answer convergence and output alignment. After fine-tuning, the model more consistently produces parsable structured final answers and shows gains on discrete judgment tasks such as `formal_fallacies`, `navigate`, `web_of_lies`, and `boolean_expressions`. Thus, A²RBench may serve not only as an evaluation benchmark but also as a training signal for answer convergence, output alignment, and discrete rule discrimination.
>
> Results at the best checkpoint (`epoch 2`) are summarized below.
>
> | Dataset | Before FT | After FT | Gain |
> |---|---:|---:|---:|
> | MMLU-Pro General | 16.35% | 34.13% | +17.78 |
> | BBH Reasoning | 9.15% | 13.24% | +4.09 |
> | formal\_fallacies | 2.8% | 59.2% | +56.4 |
> | navigate | 10.8% | 59.6% | +48.8 |
> | web\_of\_lies | 21.2% | 41.6% | +20.4 |
> | boolean\_expressions | 64.0% | 82.8% | +18.8 |
> | BBH JSON format alignment | 29.04% | 100.00% | +70.96 |
> | MMLU output-format alignment | 13.54% | 99.95% | +86.41 |
>
> # **W5. Interpreting GPT Family Differences**
>
> We added `gpt-5.4` and `gpt-5.4-mini` as rebuttal experiments on the `108`-task seed subset. `gpt-5` has the highest accuracy, followed by `gpt-5-mini`, and we suspect that one reason `gpt-5` outperforms `gpt-5.2` is its much larger token budget. Notably, `gpt-5.2` and `gpt-5.4` use nearly the same average total tokens, yet `gpt-5.4` performs better. This suggests that the difference is better explained by reasoning budget and solution strategy than by monotonic capability ordering.
>
> | Model | Accuracy | Avg Total Tokens |
> |---|---:|---:|
> | gpt-5 | 37.96% | 3175.1 |
> | gpt-5-mini | 33.33% | 2132.1 |
> | gpt-5.4 | 23.15% | 989.3 |
> | gpt-5.2 | 17.59% | 992.9 |
> | gpt-5.4-mini | 11.11% | 934.9 |
>
> We thank the reviewer for the insightful suggestions, which have helped us strengthen the paper.

---

> > ### Author Rebuttal · Reviewer_dpxU · 2026-04-05
> >
> > Thanks for your effort on the rebuttal. I'll maintain the score.

---

### Official Review · Reviewer_eqhd · 2026-03-09

**Soundness:** 3
**Presentation:** 2
**Significance:** 3
**Originality:** 3
**Overall Recommendation:** 3
**Confidence:** 3

**Summary:**

This paper introduces A²RBench, an automated pipeline for generating, expanding, evaluating, and analyzing abstract reasoning benchmarks for LLMs. To overcome the hallucination issues typical of LLM-generated datasets, the authors introduce a "cycle consistency" framework—requiring the LLM to generate both a forward function and its inverse as executable code, ensuring programmatic verification (e.g. $g(f(x)) = x$). The paper evaluates 14 LLMs across 1,054 tasks spanning 1D/2D/3D and symbolic/semantic domains. Key findings include: (1) top models significantly lag humans ; (2) a "dimensionality bottleneck" where 2D tasks are harder than 3D due to author model limitations; (3) an "augmentation paradox" where higher input complexity paradoxically improves accuracy.

**Compliance With Llm Reviewing Policy:**

Affirmed.

**Key Questions For Authors:**

1. How do you address the gap between finite testing and universal guarantees? Theorem 3.1 assumes g(f(x)) = x for all x ∈ X, but you only test on X_test. Have you observed cases where cycle consistency passed on examples but failed on the query? What proportion of tasks are tested on how many inputs?
2. How diverse are the 72 seed rules? Can you provide a clustering or taxonomy analysis showing that the rules cover meaningfully different reasoning primitives? What is the overlap or similarity between rules generated by different author models?
3. How reliable is the LLM judge? You use GPT-5-Mini for judging correctness, triviality filtering, and cognitive analysis. What is the agreement rate between the LLM judge and human annotation on a sample? Particularly for the cognitive classification (surface fitting vs. true generalization), have you validated this taxonomy?

**Limitations:**

The paper acknowledges that task complexity is bounded by the author LLM's generative ceiling, and that the cognitive analysis relies on an LLM proxy for Occam's Razor. Additional limitations not fully addressed include: the bijectivity constraint fundamentally restricts the space of expressible reasoning tasks; the human evaluation is too limited to draw strong conclusions about the human-LLM gap and how reliable  the LLM judge is lacks of testing.

**Strengths And Weaknesses:**

Strengths:
1. Principled verification framework. The cycle consistency check is an elegant and sound mechanism.
2. Cost efficiency and scalability. The seed-then-expand paradigm achieves a large cost reduction per task during expansion, making the approach highly practical.
3. Comprehensive evaluation. 14 models are tested, with detailed breakdowns by dimension, rule type, author model, and variation type.

Weaknesses:
1. Bijectivity restriction severely limits task scope. Requiring all rules to be bijections excludes many natural abstract reasoning tasks (e.g., counting, summarization, filtering). The paper claims this is "the only viable option" but this conflates verifiability with the full space of valid reasoning tasks.
2. Heavy reliance on LLM judges for non-trivial assessments. While cycle consistency is deterministic, the pipeline still uses LLM judges for: (a) filtering trivial tasks, (b) verifying code-rule consistency, (c) evaluating final answers, and (d) cognitive analysis. The paper criticizes LLM-as-judge unreliability but then uses LLM as judge throughout. No inter-annotator agreement or judge reliability analysis is provided.
3. The "augmentation paradox" explanation is underdeveloped. The claim that structured inputs reduce rule ambiguity is intuitive but not rigorously established. V4-V6 are edge cases (e.g., empty structures, minimal inputs), which are trivially easy for many transformations—this is not truly paradoxical but expected. The compression ratio metric conflates structural regularity with task difficulty.

---

> ### Author Rebuttal · Authors · 2026-03-31
>
> # **W1. Bijectivity as a Design Choice**
>
> Indeed, not all abstract reasoning problems are bijective. For our method, however, bijectivity is necessary for automated verification, because it makes cycle-consistency verification feasible and greatly improves verification feasibility. Bijectivity is therefore a trade-off between task breadth and the feasibility of automated evaluation. We also agree that our earlier wording may overclaim, and we will revise it accordingly.
>
> # **W2Q3. Reliance on LLM Judges**
>
> This is an important concern. We added human review for the first three judgment stages in the pipeline on `100` sampled items each. Using `5` independent undergraduate student annotators and comparing the human majority vote with `gpt-5-mini`, the agreement rates are `94.0%`, `97.0%`, and `100.0%`.
>
> For the cognitive taxonomy, we used the same five-annotator majority-vote protocol on the `90`-item success set before comparison with `gpt-5-mini`. Agreement is `29/31 = 0.9355` for `True Generalization`, `19/23 = 0.8261` for `Inferior Rule`, `26/36 = 0.7222` for `Surface Fitting`, and `74/90 = 0.8222` overall. The corresponding metrics are `Cohen’s κ = 0.7333` and `Macro-F1 = 0.8186`. These results agree well with human majority labels. We do not claim that the LLM judge is noise-free. In the revision, we will report these results in an appendix subsection.
>
> # **W3. Augmentation Paradox**
>
> We agree that the original explanation is not sufficiently clear. Our intended logic is: more structured inputs -> lower failure entropy -> fewer types of incorrect rules extracted by the model -> lower rule-space ambiguity -> higher solver accuracy. Appendix D.3 shows that when compression ratio rises from `1.156` at `V0` to `4.286` at `V4`, failure entropy drops from `2.825` bits to `1.532` bits. Our edge cases are not limited to empty structures or minimal inputs; more importantly, they also include cases where the input falls exactly on a rule partition boundary. By “paradox,” we mean that when the underlying rule is fixed and only the input instance changes, higher information complexity can sometimes reduce rather than increase rule-induction difficulty, contrary to the usual intuition. **In our paper, compression ratio quantifies input complexity rather than task difficulty; difficulty depends instead on rule ambiguity and the error distribution, and we will make this distinction clearer in the revision.**
>
> # **Q1. Finite Testing vs. Formal Guarantees**
>
> For each task, we perform cycle-consistency checking over the full input space, which includes both the examples and the query (see line 261 in the main text). Therefore, under the current setup, there is no case in which cycle consistency passes on the examples but fails on the query. We will clarify this in the revision.
>
> # **Q2. Diversity of the 72 Seed Rules**
>
> We agree that this point requires more direct quantitative analysis. We therefore added an embedding-based clustering analysis of the `72` seed rules. We embed each rule description with `text-embedding-3-large`, cluster them with `AgglomerativeClustering` (Ward linkage, Euclidean distance), search `k in [6,16]`, and select the best `k` by silhouette score, yielding `k=15`. All `72` rules are textually unique and spread across `15` clusters rather than collapsing into a few templates, indicating substantial diversity. Cross-author cosine similarities remain in a moderate range (`0.473-0.547`), while within-author averages remain comparable (`0.476-0.628`). The diagonal is not `1` because it excludes self-comparisons and instead measures average similarity among different rules from the same author model.
>
> | Author B / Author A | gpt-5-mini | gemini-2.5-flash | gemini-2.5-pro | o4-mini |
> |---|---:|---:|---:|---:|
> | gpt-5-mini | 0.628 | 0.546 | 0.509 | 0.547 |
> | gemini-2.5-flash | 0.546 | 0.562 | 0.513 | 0.518 |
> | gemini-2.5-pro | 0.509 | 0.513 | 0.476 | 0.473 |
> | o4-mini | 0.547 | 0.518 | 0.473 | 0.534 |
>
> We have responded point by point and added new evidence, especially the human-validation analyses. We believe the paper’s central contribution lies in an automated pipeline for generating, expanding, evaluating, and analyzing abstract-reasoning tasks, together with a cycle-consistency-based verification framework that enables scalable benchmark construction without sacrificing rigorous task validation. **In light of these clarifications and added evidence, we hope you may reconsider the Overall Recommendation.** We thank the reviewer for the thoughtful feedback.

---

### Official Review · Reviewer_YJGS · 2026-03-09

**Soundness:** 2
**Presentation:** 3
**Significance:** 3
**Originality:** 3
**Overall Recommendation:** 4
**Confidence:** 3

**Summary:**

This paper introduces A2RBench, an automated pipeline for generating abstract reasoning benchmarks to evaluate LLMs. The pipeline has four stages: (1) seed generation:"author" LLM produces bijective transformation rules implemented as forward-inverse function pairs in Python; (2) task expansion: "expander" LLM generates new inputs for validated rules; (3) evaluation:"solver" LLMs attempt to infer rules from examples and apply them to held-out queries; and (4) analysis: an "analyst" LLM classifies the quality of reasoning. The key design choice is requiring all rules to be bijective, implemented as executable forward $f$ and inverse functions $g$. This enables a cycle consistency check (verifying $g(f(x)) = x$) as an automated filter for flawed tasks. The resulting benchmark contains 1,054 tasks spanning 1D sequences, 2D grids, and 3D voxel arrays. Evaluating 14 models, the paper reports three main findings: LLMs significantly underperform humans on a representative subset, author LLMs appear to generate simpler 3D tasks than 2D tasks, and more structured inputs can paradoxically make tasks easier to solve by constraining the space of consistent rules.

**Compliance With Llm Reviewing Policy:**

Affirmed.

**Key Questions For Authors:**

Please see my discussion of weaknesses above.

**Limitations:**

Yes.

**Strengths And Weaknesses:**

Strengths:

(1) Benchmark contamination and memorization are threats to evaluating LLM reasoning. The paper's solution to build a pipeline that generates reasoning tasks with LLMs and automatically verify that the reasoning tasks are valid is a compelling idea. I found the paper's key trick (constraining tasks to bijective transformation) to be a clever choice that makes scalable generation possible. It enables cheap verification without human notation or LLM-based judging.

(2) I found the resulting evaluation to be thorough. The paper benchmarks 14 models spanning multiple families (Gemini, GPT, Claude, Qwen, DeepSeek, GLM) across a carefully stratified task set balanced by dimensionality and rule type.

(3) The paper attempts to distinguish between "surface fitting" (i.e., picking potentially complex rules to match the examples) from actual rule induction. This is a nice distinction to draw. Since the benchmark leverages known ground truth rules and prompts LLMs to produce programs summarizing their reasoning, we can actually evaluate this distinction.

(4) I found the formalization of reasoning tasks is a useful organizational contribution. The decomposition into abstraction (infer the rule from examples) and reasoning (apply the rule to a new input) provides a clean framework for thinking about what these benchmarks measure.

Weaknesses:

(1) The paper's use of theoretical results felt unnecessary and tacked on. By my reading, theorem 3.1 reduces to the observation that if $g \circ f = \text{id}_X$, then $f$ is injective. The remaining two "conditions" verified in the proof are either true by construction (consistency: outputs are computed by executing $f$) or circular (verifiability: defined as "can be checked by a deterministic procedure," then satisfied by pointing to the cycle consistency check itself). Theorem 3.2 is a textbook restatement of Bayesian inference under the Solomonoff universal prior, with no novel content.  Moreover, Theorem 3.2 is never meaningfully operationalized: since Kolmogorov complexity is uncomputable, the paper falls back on an LLM-based analyst to classify reasoning quality, which undermines the paper's central claim of replacing subjective LLM judgment with formal guarantees.

The paper's actual contribution (identifying bijective transformations as a useful class of reasoning problems that admit cheap automated verification via cycle consistency, and building a scalable pipeline around this) is genuinely appealing and stands on its own merits. Framing this practical insight as a formal theoretical framework, complete with numbered theorems and appendix proofs, is really unnecessary in my view. The paper would be strengthened by describing the design choice plainly, justifying why bijective tasks are an interesting and useful domain for reasoning evaluation, and letting the pipeline speak for itself.

(2) The claim that LLMs "significantly underperform humans" (39.8% vs. 68.5%) is a headline finding, but the paper provides almost no detail about the human evaluation. How many participants were there? What were their backgrounds (e.g., CS students, crowdworkers, domain experts)? Were there time limits? Was there inter-rater reliability? The comparison is conducted on only 108 tasks (Table 8), and without knowing who the humans are, the gap could reflect anything from "LLMs trail typical adults" to "LLMs trail a handful of PhD students in computer science." This omission is particularly problematic because the human-LLM comparison is one of the paper's headline results!

(3) The three-way classification of reasoning quality (surface fitting / inferior rule / true generalization) is performed by an analyst LLM (GPT-5-Mini). This classification drives Figure 2 and much of the discussion in Section 5.1, yet no validation is provided. How reliable is this classification? What is the agreement rate between the analyst LLM and human annotators? Without such validation, these results rest on the same unreliable foundation the paper criticizes in its introduction and uses as the motivation for its pipeline.

(4) The paper's title and framing promise formal verification, but the cycle consistency check is only run on a finite set of test inputs. For LLM-generated code with conditional branches and edge cases, passing on a handful of inputs does not guarantee correctness everywhere. What the pipeline actually performs is unit testing — useful and practical, but categorically different from formal verification. This distinction matters because the paper's core selling point is replacing unreliable heuristics with rigorous guarantees.

(5) Many natural reasoning tasks — e.g., sorting, finding extrema — are many-to-one and therefore excluded by design. The paper frames bijectivity as a *necessary* consequence of well-posedness, arguing that uniqueness eliminates one-to-many mappings and verifiability eliminates many-to-one mappings. But these other reasoning tasks feel well-posed to me. I think the authors are trying too hard to justify the specific verification method they use when generating problems. In other words, bijectivity is a nice property to have when generating problems, but I don't buy that it is a necessary condition for a reasoning task.

(6) I struggled to interpret the authors dimensionality bottleneck result. The authors acknowledge that this pattern reflects the author LLMs' "generative capability ceiling" rather than intrinsic task difficulty. Indeed, Table 8 shows human accuracy by dimension (1D: 72.2%, 2D: 63.9%, 3D: 69.4%), which actually shows a similar dip at 2D — suggesting the pattern may partly reflect intrinsic difficulty rather than anything specific to LLMs. Doesn't this suggest that using LLMs to construct evaluation tasks is problematic?

(7) The evaluation includes multiple models from the same family (e.g., GPT-5, GPT-5-Mini, GPT-5.2, GPT4o-Mini, O4-mini; Qwen3-8B, -14B, -32B), which would allow analysis of how abstract reasoning scales with model size or capability within a controlled architecture. For example, the Qwen results (32B: 19.3%, 14B: 14.2%, 8B: 0.0%) are suggestive of a threshold effect in model size, but this is never discussed. Such analysis would add considerable value given the benchmark's breadth.

---

> ### Author Rebuttal · Authors · 2026-03-31
>
> # **W1. Theoretical Framing and Design Choice**
>
> Your comments on Theorems 3.1 and 3.2 are very helpful. We will revise Theorem 3.1 and its proof accordingly. We also agree to demote Theorem 3.2 to an explanatory formula.
>
> More broadly, bijectivity is a design choice. We view abstract reasoning rules as mappings while keeping the benchmark pipeline fully automated rather than dependent on manual checking. Under this goal, validating whether LLM-generated tasks are correct and self-consistent becomes the central engineering challenge. Because bijective rules admit inverse functions, cycle-consistency checking becomes feasible. We will expand this discussion in the revision.
>
> # **W2. Human Evaluation Details**
>
> Indeed, these human-evaluation details need to be clarified. The human result is based on `20` CS PhD participants, half in North America and half in Asia. We add `20` CS undergraduate students and `20` non-CS undergraduate students, both recruited in Asia. All three groups completed the same `108` seed tasks without time limits, and we will continue extending this evaluation.
>
> | Group | Overall Accuracy | Fleiss' κ |
> |---|---:|---:|
> | CS PhD | 68.52% | 0.7221 |
> | CS Undergrad | 48.24% | 0.7808 |
> | Non-CS Undergrad | 26.57% | 0.5742 |
>
> These results show a clear stratified gap across participant backgrounds. **The core result is that the model trails this CS PhD group rather than a single uniform “human baseline.”** The Fleiss' κ values indicate that these gaps are stable and background-dependent. We will add this in the appendix.
>
> # **W3. Judge Reliability**
>
> We added a human agreement study on the `90`-item success set, labeled by `5` undergraduate student annotators and aggregated by majority vote before comparison with `gpt-5-mini`.
>
> | Category | Judge Accuracy |
> |---|---:|
> | True Generalization | 93.55% |
> | Inferior Rule | 82.61% |
> | Surface Fitting | 72.22% |
> | Overall | 82.22% |
>
> The corresponding metrics are `Cohen’s κ = 0.7333` and `Macro-F1 = 0.8186`. These results agree well with human majority labels. We do not claim that the LLM judge is noise-free. We will add this in the appendix.
>
> # **W4. Verification Scope**
>
> Thank you for identifying this important boundary. Indeed, our method does not provide universal formal verification of the generated program's logic over all possible inputs. Instead, it performs exhaustive deterministic validation over the full finite input set of each instantiated task. For each task, the set `X` consists of all examples and the query (around line 261), rather than a sampled test subset. **Thus, our guarantee is at the task-instance level, not over the program's entire semantic domain.** We will revise this wording accordingly.
>
> # **W5. Bijectivity as a Design Choice**
>
> Indeed, not all abstract reasoning problems are bijective. For our method, however, bijectivity is necessary for automated verification, because it makes cycle-consistency verification feasible and greatly improves verification feasibility. Bijectivity is therefore a trade-off between task breadth and the feasibility of automated evaluation. We also agree that our earlier wording may overclaim, and we will revise it accordingly.
>
> # **W6. Interpreting the 2D Bottleneck**
>
> Our claim, however, concerns the author model’s generative ceiling, so we conducted direct AST analysis of the generated rule code. Appendix D.2 (Table 6) shows that, for tasks authored by `o4-mini`, 2D rules are more logically complex than 3D rules, with `Nested If Depth = 2.33` vs. `1.40` and `Return Complexity = 15.67` vs. `2.00`. `Nested If Depth` measures conditional-branch depth, and `Return Complexity` measures the structural complexity of the final output expression. This suggests a trade-off: when the author model moves to 3D structure, it tends to simplify the underlying rule logic to keep the task valid under verification. Overall, this pattern is driven by the LLM author’s generative limitations.
>
> # **W7. Model Family Analysis**
>
> We fully agree that unified within-family analysis is worth conducting. In the Qwen3 series, `Reasoning_Collapse` is `99.91%`, `26.00%`, and `11.76%` for 8B, 14B, and 32B, respectively. This suggests that effective participation emerges only after crossing a basic scale threshold. By contrast, on the `108`-task seed subset, the accuracy and average total tokens are `37.96%` and `3175.1` for `gpt-5`, versus `17.59%` and `992.9` for the newer `gpt-5.2`. We suspect the lower budget contributes to the weaker `gpt-5.2` result. We will discuss these patterns more explicitly in the revision.
>
> We thank the reviewer for the constructive comments, which have improved the paper.

---

### Official Review · Reviewer_bhTL · 2026-03-23

**Soundness:** 3
**Presentation:** 3
**Significance:** 3
**Originality:** 3
**Overall Recommendation:** 5
**Confidence:** 2

**Summary:**

The paper introduces an automated benchmark pipeline for evaluating abstract reasoning in LLMs at scale without relying on costly manual annotation. It combines LLM-based task generation and expansion with a programmatic cycle-consistency verification method to ensure validity and unique solutions. Using this benchmark, the authors find that current LLMs still lag well behind humans in abstract reasoning, struggle especially with higher-dimensional 3D tasks, and sometimes reason more easily on inputs with greater information complexity.

**Compliance With Llm Reviewing Policy:**

Affirmed.

**Final Justification:**

The rebuttal fully resolves my concerns; I increased my overall score.

**Key Questions For Authors:**

Beyond only evaluation, how do we increase the abstraction capabilities of LLMs?

**Limitations:**

yes

**Strengths And Weaknesses:**

The paper is well-presented and -structured, and it appears to be sound.

Abstraction is considered to be at the core of human intelligence. So, the paper addresses a key aspect towards improving LLMs. The work presents an automated pipeline for generating, scaling, evaluating, and analyzing abstract reasoning tasks, enabling large-scale benchmark creation. Using this pipeline, the authors evaluate 14 mainstream models and provide a fine-grained analysis showing significant weaknesses in current models’ abstraction abilities beyond simple accuracy scores.

---

> ### Author Rebuttal · Authors · 2026-03-31
>
> # Beyond only evaluation, how do we increase the abstraction capabilities of LLMs?
>
> **Reply:**
>
> The question you raise is indeed important. We have examined it further through both human-annotation and fine-tuning experiments, and we believe A²RBench suggests at least three promising directions for improving LLMs' abstraction capabilities. First, cycle-consistency testing can serve as a reinforcement-learning reward signal. It is theoretically grounded, executable at the code level, and directly checks whether the generated forward/inverse code is logically complete and error-free. In our pipeline, this makes it well suited to encouraging reversible, complete, and generalizable rule representations rather than answers that merely fit a small set of examples. Put differently, it can reward models for learning rules that are structurally valid and logically complete, rather than only matching a few demonstrations at the surface level.
>
> Second, LLM-based evaluation of CoT quality may also serve as a process-level reward signal. What matters is not only whether the final answer is correct, but whether the model exhibits true generalization, matches an inferior rule, or merely surface-fits. This distinction goes to the core of abstraction: whether the model has extracted the correct and sufficiently general rule. We have also validated that this CoT-quality assessment is reasonably reliable. We used `5` independent undergraduate student annotators on the `90`-item success set, aggregated their labels by majority vote, and then compared the majority labels with `gpt-5-mini`. Overall agreement is `74/90 = 0.8222`, with `Cohen’s κ = 0.7333` and `Macro-F1 = 0.8186`. This suggests that LLM-based CoT-quality assessment is not arbitrary noise, but already shows clear agreement with human judgment, making it a plausible reward signal for process supervision or reinforcement learning.
>
> | Category | Correct Matches | Total | Accuracy |
> |---|---:|---:|---:|
> | True Generalization | 29 | 31 | 0.9355 |
> | Inferior Rule | 19 | 23 | 0.8261 |
> | Surface Fitting | 26 | 36 | 0.7222 |
> | Overall | 74 | 90 | 0.8222 |
>
> Third, our results suggest that A²RBench can serve not only as a benchmark but also as a training resource for abstraction-related capabilities. We performed LoRA fine-tuning on `Qwen3-8B` using supervised samples constructed from A²RBench problems, example input-output pairs, and answers. To avoid leakage from near-duplicate variants of the same underlying rule, we split the data by rule family rather than by random instance, yielding `673` training examples and `30` validation examples. We used `LoRA r=16, alpha=32, dropout=0.05`, `learning rate=2e-4`, and a `cosine scheduler`, and swept `1/2/4/8` epochs, with `epoch 2` performing best. We then observed clear gains on `BBH Reasoning` and especially on discrete judgment subsets such as `formal_fallacies`, `navigate`, `web_of_lies`, and `boolean_expressions`. Our analysis of the improved cases suggests that the gains come primarily from stronger discrete rule discrimination and more stable answer convergence. This kind of stable rule discrimination is itself a concrete manifestation of abstraction, because it reflects better extraction of the underlying rule from examples. More broadly, this suggests that A²RBench is useful not only for evaluating abstract reasoning, but also for training models toward stronger abstraction, more stable answer convergence, and improved output alignment.
>
> Results at the best checkpoint (`epoch 2`) are summarized below.
>
> | Dataset              | Before FT | After FT | Gain  |
> | ---------------------- | ----------- | ---------- | ------- |
> | BBH Reasoning        | 9.15%     | 13.24%   | +4.09 |
> | formal\_fallacies    | 2.8%      | 59.2%    | +56.4 |
> | navigate             | 10.8%     | 59.6%    | +48.8 |
> | web\_of\_lies        | 21.2%     | 41.6%    | +20.4 |
> | boolean\_expressions | 64.0%     | 82.8%    | +18.8 |
>
> In the revision, we will make this point explicit: the significance of A²RBench does not stop at evaluation; it also suggests concrete pathways for training models toward stronger abstraction.
>
> We thank the reviewer for raising this important question, which has helped us sharpen the paper.

---

> > ### Author Rebuttal · Reviewer_bhTL · 2026-04-04
> >
> > Fully resolved.

---

### Decision · Program_Chairs · 2026-04-30

**Decision:**

Accept (regular)

**Comment:**

Adding rigor to evaluating reasoning models, as well as to the process of generating datasets for evaluation, are some of the key challenges and hold great practical relevance today. From that perspective, I really like the work and the ideas presented in this work.  The paper does a great job of isloating what it means to evaluate absract reasoning, how to concretize it in tasks, and in turn how one can develop pipelines for automatically generating such benchmarks. This is especially useful given that we want to ensure there's no contamination of test datasets and training time, so any rigor we add to benchmark eval pipelines is a much welcome contribution to present day AI.

Reviewers agree on the strengths of the paper -- technical rigor, comprehensive evaluation and analysis, and conceptual contributions in terms of formalizing reasoning process. There were some concerns raised by the reviewers, to which the authors responded with convincing rebuttals. The paper makes strong contributions to AI community, and I recommed acceptance.